# Temporal TGF-β Supergene Family Signalling Cues Modulating Tissue Morphogenesis: Chondrogenesis within a Muscle Tissue Model?

**DOI:** 10.3390/ijms21144863

**Published:** 2020-07-09

**Authors:** Fei Xiong, Jörg Hausdorf, Thomas R. Niethammer, Volkm.ar Jansson, Roland M. Klar

**Affiliations:** Department of Orthopedics, Physical Medicine and Rehabilitation, University Hospital, LMU Munich, 81377 Munich, Germany; xiongfei188@hotmail.com (F.X.); hausdorf@aerztehaus-harlaching.de (J.H.); Thomas.niethammer@med.uni-muenchen.de (T.R.N.); volkmar.jansson@med.uni-muenchen.de (V.J.)

**Keywords:** tissue morphogenesis, temporal modulation, BMP-2, TGF-β_3_, BMP-7, chondrogenesis, morphogen combinations, muscle tissue

## Abstract

Temporal translational signalling cues modulate all forms of tissue morphogenesis. However, if the rules to obtain specific tissues rely upon specific ligands to be active or inactive, does this mean we can engineer any tissue from another? The present study focused on the temporal effect of “multiple” morphogen interactions on muscle tissue to figure out if chondrogenesis could be induced, opening up the way for new tissue models or therapies. Gene expression and histomorphometrical analysis of muscle tissue exposed to rat bone morphogenic protein 2 (rBMP-2), rat transforming growth factor beta 3 (rTGF-β_3)_, and/or rBMP-7, including different combinations applied briefly for 48 h or continuously for 30 days, revealed that a continuous rBMP-2 stimulation seems to be critical to initiate a chondrogenesis response that was limited to the first seven days of culture, but only in the absence of rBMP-7 and/or rTGF-β_3_. After day 7, unknown modulatory effects retard rBMP-2s’ effect where only through the paired-up addition of rBMP-7 and/or rTGF-β_3_ a chondrogenesis-like reaction seemed to be maintained. This new tissue model, whilst still very crude in its design, is a world-first attempt to better understand how multiple morphogens affect tissue morphogenesis with time, with our goal being to one day predict the chronological order of what signals have to be applied, when, for how long, and with which other signals to induce and maintain a desired tissue morphogenesis.

## 1. Introduction

The entire biological system is full of exact temporal-spatial cues in which surrounding matrix, multiple cytokines, growth and differentiation factors, antagonists, and modulators function at different expressional stages to initiate or suppress specific tissue morphogenesis [1,2,3,4,5]. Often, it is only something like a specific concentration gradient, during embryogenesis, which distinguishes between endochondral osteogenesis and articular chondrogenesis [6]. Though there are inherent similarities with both processes starting in osteogenesis and chondrogenesis states, it remains unclear what the sequence of signals is, which is activated, inhibited or modulated mechanically and/or biochemically, that then ultimately leads to the divergence of tissues from becoming bone or cartilage [6]. Whilst a large milieu of data exists on some of these signals, a detailed temporal signalling chronological map still eludes us. What is known is that the transforming growth factor-beta (TGF-β) superfamily are a critical player in several of these fundamental biological processes, among others [5,7].

Of the more than 30 members that have so far been identified in humans and divided into several ligand subfamilies, based on sequence similarities, including bone morphogenetic protein (BMP) and the TGF-β/activin subfamily [7,8,9], the TGF-β superfamily has been shown to be important in the formation of ectopic cartilage and bone in intramuscular sites in a diverse set of animals [10]. Whilst many protein members have been used singly to promote “articular” chondrogenesis, stimulate cellular phenotype, and product specific extracellular matrix (ECM) in tissue engineering in vitro [11]. However, the *in vivo* usage remains unresolved as these morphogens seem to have different properties the higher one goes in the phylogenetical tree of mammalian evolution [12,13]. BMP-2, TGF-β_3_, and BMP-7, also referred to as osteogenic protein-1 (OP-1), exhibit an irreplaceability in both osteogenesis and chondrogenesis [14,15] where they, for example, stimulate the synthesis of specific collagen matrix components and/or proteoglycans [16]. However, the detailed mechanisms and the association among these growth factors, in terms of temporal and spatial behaviour, are still unclear. Although it is not yet possible to provide clinical replaceable engineered cartilage [5,17,18], understanding at which time certain signals need to be present, for how long, and with which other proteins remains critical if this and other tissue types are ever to be properly regenerated [19,20].

Given that most implantation sites are tissues and not single cell-based aggregations, understanding how tissue reacts to growth factors is critical as these, in the end, regulate how morphogenesis progresses [21,22]. Thus, we have asked ourselves what would happen if more than one signal is added to a tissue. Whilst we do not yet understand the correct signalling cascade nor which molecules are needed and when during morphogenesis, we hypothesised that perhaps through multiple signals applied together briefly for 48 h, given that most signals *in vivo* are only present for short periods [23], or continuously over the entire culturing period, as ever more evidence suggests that only through continuous growth factor availability can the desired morphogenesis response be maintained [24], this might be the key to generating a superior response in tissues. It has been shown in previous studies that a combination of two morphogens, here BMP-6 with TGF-β_3_, can better direct stem cell differentiation towards hyaline chondrogenesis cytodifferentiation [20,25]. Similar studies in non-human primates and other species have even shown synergistic or modulatory effects of certain combinations of morphogens [26]. Cicione et al. [27] also demonstrated that mesenchymal stem cells (MSCs) in culture require a combination of the BMP-2, BMP-7 and a high amount of TGF-β_3_ to induce and maintain chondrogenesis. Whilst most studies still only focus on stem cells, the effect on tissue in culture remains a largely unexplored aspect, especially a multiple protein-based study is novel. The reason for using tissue over cells is that cells can lose their homeostatic state *ex vivo*, specifically the loss of essential amino acid building blocks required for complex protein synthesis and losing critical energy production requirements to fuel necessary anabolic activities to support formation of complex ECM components [28,29]. Furthermore, cells first need to develop a viable ECM environment before they can thrive and grow [30,31]. However, cells, unlike tissue, are better suited to determine specific differentiation patterns when exposed to growth factors as tissues, consisting of a heterogeneous population of cells with complex cell–cell interactions, may cause only specific cells to respond to extrinsically applied growth factor signals thereby limiting the cellular differentiation response to specific areas of the tissue [32]. On the other hand, various studies have shown that tissue can break down in vitro thereby releasing glucose, proteins but critically essential amino acids that can be utilized as energy and building blocks by resident cells to support the formation of more complex structures that single cells in vitro have difficulty forming [28,33,34]. As such, what would happen to a tissue stimulated with multiple morphogens over a specific period?

In the experimental model presented here, we tested a new type of muscle tissue model in which the aim was to determine what the effects would be on the temporal signalling cascade when rBMP-2, rTGF-β_3_ and rBMP-7, implicated in chondrogenesis related pathways, are added to muscle tissue in up to seven different combinations ranging from solo, to diverse pairs and a triplicate application all at once for 48 h or continuously over the entire culturing period of 30 days. Aggrecan was treated as the target antigen in the immunohistochemistry, as it is a key molecule in chondrogenesis [35]. Alcian blue staining was performed to indicate the presence of proteoglycans. Gene expression levels of fundamental makers implicated in chondrogenesis were also assessed, specifically *aggrecan* (*ACAN*), a member of the aggrecan/versican proteoglycan family, as it is an integral part of the extracellular matrix in cartilage and withstands compression [36], *sex-determining region Y (SRY)-box 9* (*SOX9*), as it is related to extracellular signal-regulated kinases (ERK), and Wnt pathway, which acts during chondrocyte differentiation with steroidogenic factor 1 and regulates transcription of the anti-Muellerian hormone gene [37]. Further, we assessed *collagen type II alpha 1* (*Col2α1*) as it encodes the alpha-1 chain of type II collagen, which is specific for cartilaginous tissues, including being necessary for the normal embryonic development of the skeleton, for linear growth, and for the ability of cartilage to resist compressive forces [38]. *Collagen type I alpha 1* (*Col1α1*) as it encodes the pro-alpha1 chains of type I collagen whose triple helix comprises two alpha1 chains and one alpha2 chain and is found in most connective tissues specifically bone, cornea, dermis and tendons [39]. *Collagen type X alpha 1* (*Col10α1*) as it encodes the alpha chain of type X collagen, is a product of hypertrophic chondrocytes including being localized in mineralization zones of hyaline cartilage [38]. Finally, we also assessed *alkaline phosphatase* (*ALP*) as it plays a central role in bone mineralization [40]. Ultimately, the importance of such a model cannot be underestimated, especially for clinical aspects in our field, i.e., tissue engineering. Such a model, if successful, could be used to test an unlimited amount of signals at different time intervals and limitless combinations that could shed important information in the future on the inner workings of the biochemical processes that direct tissue morphogenesis.

## 2. Results

### 2.1. qRT-PCR

#### 2.1.1. *ACAN*

Overall assessment of *ACAN* expression in the groups with continuously applied morphogen(s) for 30 days, showed that all treatment groups were significantly upregulated compared to the corresponding cultured control at each of the three culturing time points. At day 7, the expression of *ACAN* in the group with rBMP-2 was significantly upregulated over all other treatment modalities. When muscle tissue was exposed to rTGF-β_3_ + rBMP-7, the *ACAN* expression was upregulated significantly at day 14 compared to any other growth morphogen(s) treatment groups except for the rBMP-7 only group. At day 30, rBMP-2 + rTGF-β_3_ + rBMP-7 treatment was significantly greater upregulated than all other treatment groups (Figure 1, Appendix A).

The interactions between the stimulation duration (only 48 h or continuous) and the culture sampling time (7, 14, or 30 days) in different treatment modality groups were significant when analysed using two-way analysis of variance (ANOVA). Analysing the gene expression patterns over time for *ACAN*, the control group showed a slightly upregulated expression pattern. In the treatment groups with continuously applied morphogen(s), the expression of *ACAN* in the group with rBMP-2 peaked at day 7 and then severely decreased to a stable lower upregulated expression state. The expression level of *ACAN* in rBMP-2 + rTGF-β_3_ groups decreased after day 7. The rTGF-β_3_, rBMP-7, rBMP-2 + rBMP-7, and rTGF-β_3_ + rBMP-7 groups peaked on day 14 and then remained stably expressed until day 30. From day 7 to 14, rBMP-2 + rTGF-β_3_ + rBMP-7 group did not show any significant changes, however by day 30, *ACAN* expression was significantly upregulated (Figure 1, Appendix A).

During the entire culture progress, the relative expressions of *ACAN* in the groups where morphogen(s) were applied continuously were significantly higher than the corresponding 48 h withdrawal groups. None of the muscle tissue fragments treated for 48 h initially with the different growth factors and their combinations maintained a statistical difference to that of the corresponding control groups at all three culturing time points. The only notable difference in the groups where growth factors were applied for 48 h was found in rBMP-7, rBMP-2 + rTGF-β_3_, and rBMP-2 + rBMP-7 groups in which *ACAN* was significantly decreased in its upregulation by day 14, followed by a significant increase by day 30 (Figure 1, Appendix A). 

#### 2.1.2. *SOX9*

Overall assessment of *SOX9* expression in groups with continuously applied morphogen(s) showed that most treatment groups were significantly upregulated compared to the corresponding culturing control at each of the three culturing time points. Similar to the expression pattern of *ACAN*, the greatest upregulation of *SOX9* occurred in rBMP-2, rTGF-β_3_ + rBMP-7, and rBMP-2 + rTGF-β_3_ + rBMP-7 groups at day 7, 14, and 30, respectively, showing varying significances from all of the other experimental groups at their most prominent stage (Figure 2, Appendix A).

The interactions between the stimulation duration and the culture sampling time in different treatment modalities groups were significant when using two-way ANOVA. Analysing the gene expression patterns in the groups applied morphogen(s) continuously over time for *SOX9*, the relative upregulated expression level in the rBMP-2 group sharply decreased significantly from day 7 to 14, after which a stable lower upregulated expressional state was maintained. The most of other treatment groups remained stable after reaching a peak on day 7. Although rTGF-β_3_ and rTGF-β_3_ + rBMP-7 groups showed an upregulation trend from day 7 to 14, they were not significantly different (Figure 2, Appendix A). 

During the entire culture progress, the relative expressions of *SOX9* in the groups where growth factors were applied continuously over time were all significantly higher than the corresponding 48 h withdrawal groups, except for the rBMP-2, rTGF-β_3_, and rTGF-β_3_ + rBMP-7 groups at day 14. At day 30, expressions of *SOX9* in all the experimental groups where morphogen(s) were applied only for 48 h the gene expression was downregulated (Figure 2, Appendix A).

#### 2.1.3. *Col2α1*

Overall, in the assessment of *Col2α1* expression in treatment groups with continuously applied morphogen(s) over time, all experimental groups were significantly upregulated compared to the culturing control at day 7 and 14, except the rTGF-β_3_ group at day 7. At day 7, the expression of *Col2α1* in the group with rBMP-2 was significantly upregulated over all other treatment modalities. When muscle tissue was exposed to rBMP-2 + rTGF-β_3_, the *Col2α1* expression was highest at day 14 compared to any other growth factor(s) treatment groups, but there was no significant difference among the other experimental groups. At day 30, the expression of *Col2α1* could not be detected by qRT-PCR in some groups with continuously applied morphogen(s), resulting in a data volume gap for rTGF-β_3_, and rBMP-2 + rBMP-7 such that the statistical analysis was limited (Figure 3, Appendix A).

Due to the lack of data, two-way ANOVA was not applicable for analysing the interaction between the stimulation duration and the culture sampling time in rTGF-β_3_ and rBMP-2 + rBMP-7 groups, and one-way ANOVA and multiple testing correction were used to analyse these two factors in the before mentioned groups. The interactions between these two factors in the other experimental groups were significant. Analysing the gene expression patterns over time for *Col2α1*, the pattern in most groups stimulated continuously showed a decreasing or stable state from day 7. The group with continuously applied rBMP-2 showed a sharply significant decrease in the upregulation pattern from day 7 to 14 (Figure 3, Appendix A). 

During the entire culture progress, the relative expression of *Col2α1* in most groups where growth factors were applied continuously over time was significantly higher than the corresponding 48 h withdrawal groups. Overall assessment of *Col2α1* expression in treatment groups applied for only 48 h showed that the relative expression levels in all the experimental groups were downregulated from day 14, except rBMP-2 + rBMP-7 and rBMP-2 + rTGF-β_3_ + rBMP-7 groups that were upregulated at day 30. At day 30, the *Col2α1* could be detected by qRT-PCR in all groups in which morphogen(s) were applied only for 48 h (Figure 3, Appendix A).

#### 2.1.4. *Col1α1*

Overall assessment of *Col1α1* expression in groups where morphogen(s) were applied continuously over time showed that the expression levels in all treatment groups was significantly downregulated at day 7 and 14 versus the corresponding cultured control group, except the rBMP-7 group on day14 that showed downregulation. The rTGF-β_3_ + rBMP-7 and rBMP-2 + rTGF-β_3_ + rBMP-7 group showed no difference at day 30 including the rTGF-β_3_ group that at day 30 was higher upregulated (Figure 4, Appendix A). 

The interactions between the stimulation duration and the culture sampling time in different treatment modalities groups were significant when analysed using a two-way ANOVA. Unlike the pattern of the other genes, the relative expressions level of *Col1α1* in the treatment groups with continuously applied morphogen(s) were all significantly lower compared to the 48 h single stimulation groups over most of the three culturing time points (Appendix A).

The upregulation of *Col1α1* over the entire culturing period from day 7 to day 30 was found only in the culturing controls and the treatment group with 48 h applied rTGF-β_3_. The relative expression levels in the other experimental groups, irrespective of whether morphogen(s) were applied for 48 h or continuously, were highly downregulated at day 7 followed by a significant gradual increase in the expression pattern over time with *Col1α1* being slightly upregulated by day 30. The expression was where *Col1α1* was still downregulated by day 30 was in the continuously applied rBMP-2 + rBMP-7 group (Figure 4, Appendix A).

#### 2.1.5. *Col10α1*

Although mRNA was extracted from six samples in each experimental group with continuously applied morphogen(s), the expression of *Col10α1* could not be detected. Similarly, the expression of *Col10α1* in groups with 48 h applied rTGF-β_3_ + rBMP-7 and rBMP-2 + rTGF-β_3_ + rBMP-7 could not be detected at day 7 but was recovered at day 14 and 30. The relative expression levels of *Col10α1* in groups exposed to a 48 h single stimulation was downregulated at all three culturing time points. The expression in most treatment groups showed no difference compared to the cultured control group over all three culturing time points, except for rBMP-7 group at day 7 and rBMP-2 + rBMP-7 group at day 30 (Figure 5, Appendix A). Two-way ANOVA was not performed when analysing the expression of *Col10α1*, due to the lack of data in some groups.

#### 2.1.6. *ALP*

The relative expression of *ALP* was only upregulated in the group with continuously applied rBMP-2 at day 7. After day 7 and continuing to day 30, the relative expressions of *ALP* in both rBMP-2 and all other groups irrespective of morphogen application length showed a downregulated state. The interactions between the stimulation duration and the culture sampling time in most treatment modalities groups were significant except in the rBMP-7 treated group when analysed using a two-way ANOVA. The main effects of these two factors were also not significant in rBMP-7 treated group. Interestingly, the pattern of the continuous stimulation was significantly higher than the corresponding 48 h continuously applied until day 14 with results being reversed between the experimental groups at day 30 yet only so long as rBMP-2 was involved (Figure 6, Appendix A).

### 2.2. Histologic and Immunohistochemical Analysis 

#### 2.2.1. Alcian Blue Staining

Alcian blue staining confirmed chondrogenic morphogenesis under our experimental conditions. Alcian blue showed the presence of glycosaminoglycans in cartilages as well as other body structures, and the positive area was stained in blue. The positive area was found in nearly all the groups at all time points. The positive areas were mainly in the intercellular space of muscle cells or near the fascia (Figure 7).

Histomorphometrical analysis of alcian blue staining showed that the positive area ratios were lower than 5% at three culturing time points in the control group, although it presented an increasing trend. The range of positive area ratios at three culturing time points in single and continuous stimulation experimental groups were 0.33–10.57% and 0.48–41.72%, respectively. Overall assessment of the positive area ratio in treatment groups with continuously applied morphogens over time showed that all treatment groups were significantly higher compared to the corresponding control at each of the three culturing time points. The highest ratio was found in the group treated continuously with rBMP-2 + rBMP-7 at day 14. The interactions between the stimulation duration and the culture sampling time in most treatment modalities groups were significant except in the rBMP-2 + rTGF-β_3_ treated group when analysed using two-way ANOVA. The main effects of these two factors were significant in rBMP-2 + rTGF-β_3_ treated group. The positive area ratio in most groups where growth factors were applied continuously over time were all significantly higher than the corresponding 48 h withdrawal groups at day 14 and 30 but showed no difference at day 7 except rBMP-2 + rBMP-7 groups (Figure 8, Appendix A).

#### 2.2.2. Aggrecan Immunohistochemistry

By means of immunohistochemistry, in which aggrecan was treated as the antigen, the green area indicated the positive antigen-antibody interactions. Whilst the positive area could be detected in nearly all the groups at all culturing time points, the control and the 48 h single stimulation groups showed a very weak positive reaction. The areas that reacted positively to the immunohistochemical staining occurred throughout the tissue section including the muscle cells, the intercellular space and the fascia (Figure 7).

Tissue sections from continuously applied rBMP-2 showed the highest antigen density among all groups at day 14 and then decreased sharply and significantly, which was similar to the relative chondrogenesis-related genes expression patterns from day 7 to 14. Additionally, the strongest positive reaction at day 30 was found in continuously applied rBMP-2 + rTGF-β_3_ group, which although not significantly different to the continuously applied rTGF-β_3_ + rBMP-7 group, the latter group interestingly presented the greatest chondrogenesis-related genes expression at day14 (Figure 9, Appendix A).

The interactions between the stimulation duration and the culture sampling time in most treatment modalities groups were significant except in the rTGF-β_3_, rBMP-7 and rBMP-2 + rTGF-β_3_ + rBMP-7 treated groups when analysed using a two-way ANOVA. The main effects of these two factors were significant in rBMP-2 + rTGF-β_3_ + rBMP-7 treated group, while the main effect was only significant for stimulation duration in rTGF-β_3_ and rBMP-7 groups.

### 2.3. Hierarchical Clustering

At each time point, 15 groups were analysed using hierarchical clustering and each group contained 6 replicate samples. At day 7 (Figure 10A), the gene expression in the group with continuously applied rBMP-2 was different from the other groups. Additionally, the expression of *Col2α1* to *ALP*, as well as *ACAN* to *SOX9* showed a close relationship to one another between all the samples and were highly upregulated in the muscle tissue to which rBMP-2 was applied continuously. At day 14 (Figure 10B), two final clusters developed in which muscle tissue groups separated into continuously (Cluster 2) and 48 h (Cluster 1) applied morphogenes with Cluster 1 also containing the control groups. The expression of *Col2α1* and *ACAN* were closely related to each other considering all the samples. At day 30 (Figure 10C), the previously two identified clusters that formed for day 14 were maintained, with the expression of *SOX9* and *ACAN* now showing a close relationship to each other when considering all the samples. The expression of *ALP*, *Col1α1* and *Col10α1* were highly upregulated in the groups with 48 h applied rBMP-2 + rBMP-7 and rBMP-2 + rTGF-β_3_ + rBMP-7 (Figure 10).

The results for the percentage of the positive alcian blue staining area, under hierarchical clustering (Figure 11), combined all three culturing time periods instead of separating them as was done for the gene clustering. For the alcian blue clustering the groups with continuously applied rBMP-7 and rBMP-2 + rBMP-7 showed that the percentage of the positive alcian blue staining area was very high at day 14 (Cluser 2) which was maintained until day 30 (Cluster 2). This was not the case with the other groups. Additionally, all the groups exposed to morphogens continuously showed a high percentage of positive alcian blue staining area at day 30 which was different from the single stimulation groups containing the control group (Figure 11).

## 3. Discussion

Members of TGF-β superfamily are well known for their functions in nearly every aspect of development, from early vertebrate growth to postnatal life [4,5,41,42]. Johnstone et al. [43] first described chondrogenesis cultures in defined media by adding TGF-β_1_ and dexamethasone in 1998. Over the past decades, much more emphasis has been given to the role of the TGF-β superfamily in cartilage formation. Whilst many studies have focused on the effects of single growth factor applications on mesenchymal stem cytodifferentiation, little research has yet investigated the impact of what different combinations of these factors do in vitro especially on tissues. In this study, we analysed the in vitro chondrogenic morphogenesis at three time points in muscle tissue exposed to three growth factors (rBMP-2, rTGF-β_3_ and rBMP-7) alone, in a pairwise and even triplicate combination for only 48 h or continuously for 30 days, in which, as is the normal state *in vivo*, a constant exposure of key initiators and subsequent follow up modulators ultimately drive proper tissue transformation was considered.

Immunohistochemistry and alcian blue staining results in our study revealed that growth factors induced a form of chondrogenesis resulting in the formation of specific ECM. This remains a critical aspect in tissue engineering approaches where chondrogenesis in vitro is distinguished by the type of matrix proteins that are expressed/formed. Is the ECM hyaline or does it contain significant amounts of type I collagen, and if it is a hyaline matrix does it develop towards stable articular cartilage or rather progresses towards hypertrophy and mineralization? Whilst other markers have also been identified that seem to be unique for articular chondrogenesis [44], classical markers remain critical for the final validation [45]. *ACAN*, *SOX9,* and *Col2α1* are markers that are generally utilised to monitor if any cartilage formation had occurred irrelevant of the type [46]. *Col1α1* and *Col10α1,* on the other hand, are classical markers to further differentiate what type of chondrogenesis is underway [47]. Articular chondrogenesis generates a matrix that has superior load-bearing and mechanical properties and is free of type I collagen [48], while hyaline chondrogenesis formed during endochondral ossification of embryogenesis for certain skeletal bones, is characterized by the early appearance of type X collagen [48]. The presence of both type I and II collagen, on the other hand, is indicative of fibro-chondrogenesis [48]. Several studies have shown that proteoglycans do not only accumulate in cells but are also in vitro in the medium when cells are exposed to only TGF-β_3_ [27,49]. *SOX9* expression is an essential factor for the initiation of chondrogenesis [50], as it promotes the transcriptional program and drives the upregulations of *Col2α1* and *ACAN* [51,52,53]. Yoon and Lyons [54] reviewed that *SOX9* as the transcription factor was continuously expressed in chondrocytes up to the hypertrophic stage, where BMPs were required. In relation to the present study, both the positive reaction of immunohistochemistry/histology staining and the significantly higher expression of *ACAN* and *SOX9* genes in the various treatment groups indicated a trend towards a chondrogenic morphogenesis.

From day 14, the density of aggrecan, in the immunohistochemistry, in most continuous application groups were significantly higher than in the corresponding 48 h stimulation groups, which suggested that short-term stimulation is insufficient to initiate proper morphogenesis in the muscle tissue. The relative expression levels of *ACAN*, *SOX9*, and *Col2α1* including the hierarchical clustering at day 14 and 30, supported this conclusion. Jelic et al. [55] demonstrated that prolonged administration of BMP-7 showed better regeneration of articular cartilage chondral defects compared to a single intra-articular injection. This would suggest that the creation of a morphogenic tissue response is dependent on the sustained stimulation and not a temporally limited activation period. However, *in vivo* experiments performed by Neol et al. [56] revealed that short-term *BMP-2* expression was sufficient to induce the osteochondral differentiation by Tet-Off system. This makes it difficult to interpret the results from previous research groups as it would seem that certain morphogenic events have different growth factors present at specific time intervals that may or may not be consistent between *in vivo* and in vitro techniques [57]. Alternatively, the inconstancy in previous studies could also lie in the different delivery mechanisms in morphogen application. For example, in the present study, 48 h stimulation was merely affected by the release kinetics that did not reach a proper concentration at the target site, thereby causing a limited response. This and the vast amount of variables between these experimental types, including the choices of cells versus tissue in the presence of various morphogens, are all criteria that have to be considered in the present experiment making this a daunting task to interpret.

Toh et al. [58] initially demonstrated that BMPs could induce the formation of bone and cartilage in ectopic sites, acting as an autocrine and/or paracrine regulation factor that modulates the development of bone and cartilage. Whilst they described BMP-2 regulatory effect acting on the supposed maturation of mesenchymal progenitors where it promotes the synthesis of chondrocyte matrix formationl [58], Chan et al. [59] showed that BMP-2 locally affects all cell types within the ectopic muscle site including stem cells that can be directed either towards an osteogenesis or chondrogenesis pathway based on the availability of vascular endothelial growth factor (VEGF). In addition, by comparing the different effects of BMP-2, -4, and -6, Sekiya et al. [60] also suggested using BMP-2 to produce high quantities of polysaccharide-rich cartilage, with Schmal et al. [61] further demonstrating the importance of BMP-2 in cartilage repair and maintenance through in vivo experiments. However, since BMP-2 is more centrally directed to induce endochondral bone development [62,63], our results, in conjunction with previous discoveries, would suggest that BMP-2 is important as an initiator of endochondral bone chondrogenesis that can only be maintained or altered provided specific other growth factors are present at the correct time, which then define the direction the tissue develops into [59,64]. Indeed, at day 14, the group with continuously applied rBMP-2 only, in the present study, showed the highest density of aggrecan antigen, which then dropped sharply. This was consistent with the expression results of *ACAN*, *SOX9*, and *Col2α1* from day 7, supported by hierarchical clustering. Van Beuningen et al. [57], when injecting murine knee joints with BMP-2 or TGF-β_1_ over consecutive days, showed that BMP-2 stimulated chondrocyte proteoglycan synthesis sooner and stronger than TGF-β_1_, but the stimulation duration of BMP-2 was shorter. Hellingman et al. [65] indicated that phosphorylation of both SMAD2/3 and SMAD1/5/8, stimulated by TGF-β and BMP subfamily ligands respectively, are essential at the initial activation of chondrogenic differentiation where these SMADs stay active in differentiated MSCs, with only SMAD2/3 being present in native articular chondrogenesis. This suggests that induced chondrogenesis can only be maintained through the regulatory effect involving SMAD2/3 regulated by the TGF-β subfamily protein(s). On the other hand, BMP-7 upregulates chondrocyte metabolism, stimulates not only cartilage-specific extracellular proteins but also generates normal functional proteoglycans [66,67,68]. This may explain the results of the present study where the higher density of aggrecan was detected in groups with continuously applied rTGF-β_3_ + rBMP-7 at day 30 including the stronger expression of *ACAN* and *SOX9* at day 14 and 30 in groups with rBMP-7 + rTGF-β_3_ and rBMP-2 + rTGF-β_3_ + rBMP-7, respectively. By analysing the results of gene expressions over time, in the different experimental groups, the extreme decline from day 7 to 14 in the group using continuously applied rBMP-2 solo was prevented by the different combinations with the other growth factors. Therefore, rBMP-2 alone could be essential to initiate chondrogenesis, yet to achieve the appropriate tissue morphogenesis, such as articular chondrogenesis, in our muscle tissue model, the correct temporal availability of rTGF-β_3_ and rBMP-7 may also be crucial.

Interestingly, the increase of density of aggrecan and the up-regulation of genes was most affected by rBMP-2 when applied solo continuously for up to 7 to 14 days. However, this effect was significantly inhibited when rBMP-2 was combined with another growth factor, where one possible reason for this could be that the TGF-β superfamily signal pathways have been postulated to antagonize each other, with the exact mechanisms not yet clearly understood [4,69]. The functions of TGF-β ligands in chondrogenesis, as reviewed by Wang et al. [5], show that TGF-βs can participate in both TGF-β isoform (SMAD2/3) and BMP isoform (SMAD1/5/8) signals, but which TGF-β isoform utilises which BMP receptor remains unknown. Gronroos et al. [70] has shown that TGF-β isoforms can inhibit BMP-induced transcription by forming the inhibitory complex SMAD1/5-SMAD3 that prevents proper receptor activation. Since, in the present results, similar antagonistic patterns were observed between the three growth factors when these were combined with each other during culturing, especially during the early stages of culture at day 7, something similar may have happened and may need future studies to clarify. Whilst we have seen a clear pattern of rBMP-2 upregulating all genes in the absence of the other morphogens, yet losing its effectiveness prior to day 14, the question to pose is what signal(s) are necessary next in the cascade? Since the other growth factors retarded rBMP-2 effect and only minimally affected gene expression patterns at the later stages of culture, even when these were in the presence of rBMP-2, are these growth factors truly critical during day 14 to 30 or is there something missing? We cannot clearly answer this when more research definitely has to be conducted to understand the exact sequence of the signals and what quantity is critical to ensure a properly sustained reaction. 

The polymeric extracellular framework is formed by collagen in almost all animal tissues [71,72]. The relative expression of *Col1α1*, *Col2α1*, and *Col10α1*, whose encoded proteins are components of the ECM of fibrocartilage, hyaline cartilage, and hypertrophic cartilage, respectively [73,74], were all detected by qRT-PCR in this study. The co-delivery of TGF-β_3_ and BMP-2 into MSCs performed by Gonzalez-Fernandez et al. [62] suppressed hypertrophy and produced a more stable chondrogenesis, whereas Cals et al. [75] did not find any significant TGF-β subtype-dependent differences in the expression of *Col10α1*. Other studies further demonstrated an increase of *Col10α1* expression during the chondrogenesis of MSCs without the application of BMPs [43,76]. As such, the absence of *Col10α1* expression in nearly all experimental group under continuous stimulation may indicate that these three growth factors could not induce chondrocyte hypertrophy or pure endochondral ossification within the muscle tissue. Additionally, the absence of the qRT-PCR results in some experimental group when amplifying *Col2α1* cDNA made our analysis incomplete, which may have been caused by the destabilization during the translation of some mRNA with AU-rich elements [77,78,79]. More efforts need to be done to explore the convincing reason of the absence and recovery of these two genes in further studies. The downregulation of *Col1α1* at day 7 and 14 in most of our experimental groups, in relation to the *Col2α1* expression, suggested that our induction was not directed towards fibro-chondrogenesis indicative by its type I and II collagen content [48]. While analysing the time-dependent synthesis of chondrogenesis-specific ECM protein in vitro, Wang et al. [74] revealed that chondrocytes predominantly expressed type I collagen at the initial stage of culture (1-3 days), exceeding type II collagen. However, it is unclear whether the expression of *Col1α1* peaked before day 7 in our culture system due to the limitation of detection time. Ninty-95% collagen produced by chondrocytes is type II collagen, which is the sought after ECM material of tissue engineering when repairing the articular cartilage. However, more specific articular chondrogenesis biomarkers such as *ABI Family Member 3 Binding Protein* (*ABI3BP*), *thrombospondin 4* (*THBS4*) and *SIX Homeobox 1* (*SIX1*) seem to be required to properly specify if chondrogenesis is articular or not [44]. As such, whilst the present results at day 7 and 14 showed that, in all likelihood, the muscle tissue was stimulated and tending towards chondrogenesis, if the process was articular cannot be defined at present. However, according to the available data, the gene expression cascade analysis revealed that at day 30 the tissue irrespective of treatment with expanded culturing time the reactions was slowly tending towards fibrosis, as *Col1α1* expression seemed to be progressing towards an upregulated state occurring after the 30 day culturing cut of point. Only the rBMP-7 when applied continuously in the experimental group, seemed to negate this process of *Col1α1* upregulation by day 30 and maintained a significantly higher expression level of *Col2α1*, suggesting that rBMP-7 has a key role to play in inducing and maintaining possible hyaline chondrogenesis by preventing type I collagen production [77,78,79].

Given that this is a novel model where muscle tissue has never been conceived to ever undergo the process of chondrogenesis in vitro we did miss some critical aspects that will need to be considered in follow up experiments. Firstly, whilst we did include *ALP* as an early marker for possible osteogenesis [80], more genes and proteins have to be included especially more hyaline specific chondrogenesis markers, *ABI3BP*, *THBS4* and *SIX* [44], and the whole spectrum of both osteogenesis and chondrogenesis related pathways. This will help to better distinguish the precise chondrogenesis or other process that are being stimulated by the growth factor combinations with time. Whilst the upregulation of *ALP* did appear in the group in which rBMP-2 was applied continuously until day 7, which is also the time point that showed the most significant chondrogenesis stimulatory effect, the expressions of *Col1α1* gradually recovered after day 7, suggesting a deviation in the chondrogenesis pathway in certain groups towards endochondral osteogenesis as it occurs embryonically. However, since *Col10α1* was extremely downregulated in all the groups, in our opinion, it makes this theory improbable, as it is most likely this is perhaps two separate and unique events we cannot define clearly at the moment. Given that there are diverse cell types in muscle tissue that could be responding differently to the same stimulation [59,64], the result could also be as a consequence of specific intrinsic parameters being lost that extrinsically could be causing the tissue to behave differently even though it is exposed to similar responses faced *in vivo* [21]. As such, whether indeed all cell variants in the muscle tissue responded, as has been shown previously *in vivo* or in cell-only based experiments [59,64,81] requires more research. For this, a larger spectrum of genes, especially osteogenesis-related genes compared to well-known chondrogenesis markers, including, a limitation particularly in the present study, their translational expression patterns, via immunohistochemistry, need to be properly elucidated thereby providing key information of the exact intrinsic processes of *ex vivo* tissue morphogenesis when exposed to temporal signaling cues.

Moreover, our experiment did not carry out concentration gradient related analysis. We chose a 1:1 ratio at present at an overall dosage of 50 ng/L as in cell culture this amount has shown to promote a response that does not incur inhibitions either because of two high or low dosages. There has always been a clear evidence that has shown the biphasic effects of TGF-β superfamily members on DNA synthesis [82]. Yang et al. [83] demonstrated that the osteogenic differentiation induced by TGF-β activated kinase 1 in MSCs served as a double-edged sword, which was regulated by different concentrations of BMP-2. The expression of *Col1α1* and *Col10α1* are related to SMAD7, an intracellular inhibitor of both BMP and TGF-β isoform signalling pathway, which affects the ossification and fibrosis process [84,85,86]. Therefore, we cannot accurately conclude that the suppressed results of *Col1α*1, *Col10α1,* and even *ALP* expression was due to the participation of growth factor or an inappropriate concentration.

Finally, the chondrogenesis effect we observed in our study may not be wholly dependent on the application of growth factors. Using the gene expression level of fresh muscle tissue as the baseline, we obtained the CNRQ values of each experimental group and control group by qRT-PCR revealing changes in genes of interest [87]. The expression of *ACAN* and *SOX9* in the control group was up-regulated compared to fresh tissue (day 0, baseline), which indicated that other factors affected chondrogenesis beside the growth factors possibly due to fetal bovine calf serum used in the medium preparation. On the other hand, Fahlgren et al. [88] demonstrated that the expression of *BMP-7* increased after capsular incision. Several studies also have shown that micro-fractures possess a chondro-protective effect and stimulate cartilage repair [89,90,91]. Therefore, we have to consider whether the traumatisation of the muscle tissue, in the present study, using biopsy punches during the sample preparation process induced a tissue response that affected the gene expression. As such future research using our model may also need to consider the implication of what the mechanical [92] stimulation has on biological response as it is also part of the ligand theory and can be an important factor in determining the chronological order of the temporal signalling patterns [19,93,94,95].

## 4. Materials and Methods 

### 4.1. Acquisition of Samples

Four F-344 adult male rats (Charles River Wiga, Sulzbach, Germany) with a mean weight of 300 g were used in this study. Animals were sacrificed with an overdose of isoflurane (Abbot, Chicago, USA). Tissue harvest procedures were conducted according to the rules and regulations of the Animal Protection Laboratory Animal Regulations (2013), European Directive 2010/63/EU and approved by the Animal ethics research committee of the Ludwig-Maximillians-Universität München, Bavaria, Germany Tierschutzgesetz §1/§4/§17 (https://www.gesetze-im-internet.de/tierschg/TierSchG.pdf) with respect to animal usage for pure tissue or organ harvest only.

Under sterile conditions, a fresh abdominal muscle tissue slab was harvested and then washed with phosphate buffered saline-Dulbecco (PBS, Biochrom GmbH, Berlin, Germany) twice. The tissue was then immersed in the Alpha medium (Biochrom GmbH) containing high concentrations of penicillin and streptomycin (2%, *p*/S, Biochrom GmbH) for no more than 30 min and then washed again with PBS before being transferred to Alpha medium with 1% *p*/S. Utilising 4 mm biopsy punches (PFM medical, Cologne, Germany), in a total of 576 biopsies, equally sized circular muscle tissue fragments were harvested, treated as muscle tissue model, and placed in 96-well Nunc culture plates (Thermo Fisher Scientific, Waltham, MA, USA), in recovery medium (Alpha medium supplemented with 15% fetal bovine serum (FBS; Biochrom GmbH), 0.02 mM/mL L-glutamine (Biochrom GmbH), and 1% *p*/S for 48 h. 

### 4.2. Culture of the Muscle Tissues Model

After allowing the tissue fragments to recover for 48 h in the recovery medium, muscle tissue fragments were divided equally into a 48 h-withdrawal study (*n* = 288) and continuous 30 days (*n* = 288) morphogens application experiment, in which the medium had one of seven supplemented growth factor modalities as follows:Alpha medium supplemented with 15% FBS, 0.02 mM/mL L-Glutamine, 1% *p*/S (Normal medium-Culturing Control)Normal medium with 50 ng/mL recombinant rat (r) BMP-2 (CUSABIO, USA)Normal medium with 50 ng/mL rTGF-β_3_ (Cloud-Clone Corp., USA)Normal medium with 50 ng/mL rBMP-7 (Cloud-Clone Corp., USA)Normal medium with 50 ng/mL rBMP-2 + 50 ng/mL rTGF-β_3_Normal medium with 50 ng/mL rBMP-2 + 50 ng/mL rBMP-7Normal medium with 50 ng/mL rTGF-β_3_ + 50 ng/mL rBMP-7Normal medium with 50 ng/mL rBMP-2 + 50 ng/mL rBMP-7 + 50 ng/mL rTGF-β_3_

Recombinant rat proteins where chosen to maintain species specificity as previous research has shown that especially human TGF- β_3_ is phylogenetically different than rat TGF- β_3_ in its function to induce tissue morphogenesis [12,96]. Each modality had 36 samples that were divided up into quantitative gene (*n* = 18) as well as histological (*n* = 18) assessment groups and further into subsequent culture period lengths of 7, 14, and 30 days. In the end for each treatment modality there were always 6 muscle fragments for a given culture length and assessment method. The change from recovery medium into the modified medium was taken as day 0. The culture medium was changed every two days. Samples were cultured at 37 °C in a 5% CO_2_ humidified incubator and harvested at day 7, 14 and 30.

### 4.3. qRT-PCR

The whole qRT-PCR process was compliant with the minimum information for publication of quantitative real-time PCR experiments (MIQE) guidelines [97]. Muscle tissue fragments designated for quantitative gene analysis were (*n* = 288), once the designated culture length had been reached (7, 14, and 30 days), harvested and immediately frozen in liquid nitrogen after which they were stored at −80 °C. Under RNase-free conditions, frozen samples were ground to powders using a mortar and pestle in liquid nitrogen. Total RNA was then extracted from the specimens using the RNeasy^®®^ Fibrous Tissue Mini Kit (Qiagen, Hilden, Germany). The concentration and quality of RNA were measured spectrophotometrically using a NanoDropTMLite (Thermo Fisher Scientific)). The range of concentration of messenger ribonucleic acid (mRNA) was 89.6–125.8 ng/μL in a total volume of 30 μL. The range of A260 was 2.235–3.867. The range of A260/A280 was 1.87–2.05.

Complementary DNA (cDNA) was then reverse transcribed using a QuantiTect Reverse Transcription cDNA Synthesis Kit (Qiagen). cDNA was stored at −20 °C until used.

Quantitative RT-PCR was then performed on a qRT-PCR LightCycler^®®^ 96 Instrument (Roche, Basel, Swiss) in duplicate using 2x FastStart Essential DNA Green Master (Roche). The process of thermocycling included a 2 min denaturation step at 94 °C, 40 cycles containing a denaturation, annealing and extension step set at 95 °C for 10 s, 60 °C for 15 s and 72 °C for 30 s, respectively, and a final extension at 72 °C for 5 min. The total volume of each reaction was 10 μL, containing 5 μL Green Master, 0.6 μL forward primer, 0.6 μL reverse primer (Table 1), 2μL cDNA (10 ng in total) and 1.8 μL RNase-free water. Primers were designed utilising PrimeQuest in conjunction with OligoAnalyzer 3.1 on the IDT website (https://eu.idtdna.com/site) and optimized according to the MIQE guidelines [98,99]. GeNorm (http://medgen.ugent.be/wjvdesomp/genorm/) assessment determined that TATA-binding protein (*TBP*), glyceraldehyde 3-phosphate dehydrogenase (*GAPDH*), RNA polymerase II subunit e (*POLR2e*), ribosomal protein lateral stalk subunit P0 (*RPLP0*), succinate dehydrogenase complex flavoprotein subunit A (*SDHA*), and ribosomal protein L13α (*RPL13α*), were the appropriate reference genes to use in this experiment. Genes that were of interests included *ACAN*, *SOX9*, *Col1α1*, *Col2α1*, *Col10α1*, and *ALP*.

Sequence amplification specificity of all genes was confirmed through Sanger sequencing (GATC Biotech, Cologne, Germany) in conjunction with nucleotide mega-blast analysis (https://blast.ncbi.nlm.nih.gov/Blast.cgi?PAGE_TYPE = BlastSearch) [87,100]. Gene expression results were represented as the mean calibrated normalized relative quantities (CNRQs) and were calculated by qBase analysis software version 3.0 (http://www.biogazelle.com), which reflects the relative expression of each gene. Fresh muscle tissue was also harvested and acted as the normalization control in qRT-PCR to which all specimens were compared to.

### 4.4. Histological and Histomorphometrical Analyses–Alcian Blue

The tissue fragments to be analysed through histological processes (*n* = 288), upon harvest after being cultured for 7, 14, and 30 days were fixed in 30% formalin (Microcos GmbH, Munich, Germany) for 24 h. Specimens were then dehydrated in a Tissue Processor STP 120 unit (Thomas-medical e.U., Maishofen, Austria) and processed for paraffin wax embedding. For chondrogenesis evaluation, 2-μm thick paraffin wax sections of specimens were mounted on Superfrost glass slides (Menzel, Braunschweig, Germany), stained with alcian blue (pH 2.5, Morphisto GmbH, Frankfurt, Germany), and counterstained with haematoxylin (Morphisto GmbH) for histological and histomorphometric analyses [101,102]. Histological sections images were captured at  ×40 magnification and digitalized using a PreciPoint M8 research microscope (PreciPoint, Freising, Germany) and Viewpoint software (PreciPoint). 

Histomorphometric analysis was then performed using the Image J (https://imagej.nih.gov/ij/). The ratios of the area of positive alcian blue staining and the total area of the muscle tissue were established with values expressed as a mean percentage of 6 samples [87,100]. We used the specific RGB ranges to select the target color in histogram-based mode in Image J to reduce the variance in the selection of the positive area of the different figures. The RGB ranges of our positive area were R: 113-233, G: 156-244 and B: 195-245.

### 4.5. Immuno-Histochemical and -Histomorphometrical Analyses 

To evaluate new chondrogenesis within the muscle tissue fragments (*n* = 288), 2-μm thick paraffin wax sections were incubated with primary antibody to detect the presence of aggrecan. The primary aggrecan antibody (Biorbyt, Eching, Germany) was diluted by antibody diluent (ZYTOMED SYSTEMS GmbH, Berlin, Germany) at the concentration of 1:150. The Vina Green^TM^ Chromogen Kit (Biocare Medical) was then used to show the antigen-antibody interactions, in which a positive reaction generated a green signal. The negative controls were established by using the antibody diluent without any primary aggrecan antibody. Immunohistochemically stained sections were analysed using a PreciPoint M8 microscope and images captured at × 40 magnification using Viewpoint software. 

Histomorphometrical analysis was then performed using Image J. Firstly, the absorbance value of the incident light in the blank surrounding area of the tissue piece was calibrated after which the area of total tissue and the integrated optical density value (IOD) of positive area were measured. The mean optical density value (MOD) of the positive area was then calculated (MOD= IOD/ area), which quantified the density of immunostaining and values were expressed as a mean percentage of 6 samples. 

### 4.6. Hierarchical Clustering

Data was compiled from the various treatment modalities and stimulation durations for each of the culturing time point. Hierarchical clustering and the generated heatmaps were created using R-studio (R Studio, Boston, MA, USA, http://www.rstudio.com). The distance measure used in the clustering rows and the column was Euclidean whereas the agglomeration method used to create the clusters was based on the average. The heatmap was divided into 2 final clusters. 

### 4.7. Statistical Analysis 

Statistical assessment was performed on qRT-PCR and histomorphometric data of both histological and immunohistochemistry material using GraphPad Prism (version 7.0, GraphPad Software, San Diego, USA). A one-way ANOVA with Dunnet’s test was used to determine the statistical differences between different experimental and corresponding control groups. A one-way ANOVA with Tukey’s multiple comparisons test was used to compare the mean of each group with the mean of every other group. A two-way ANOVA with multiple comparisons was mainly used to compare the difference between each experimental group with a single and continuous application at the same time point and the changes of each experimental group over time, simultaneously. Statistical significance was defined at *p* < 0.05.

## 5. Conclusions

Properly defining the temporal signalling sequence during tissue morphogenesis could provide novel means of better promoting the desired tissue formation or could be used in ex vivo tissue engineering processes to alter any other tissue type into what is desired. In the present study, a new muscle tissue-based model tested this hypothesis utilising three well known growth factors, namely rBMP-2, rTGF-β_3_, and rBMP-7 implicated in endochondral osteo-and general chondrogenesis. By applying these solo, in pairs or as a final triplicate combination over different application durations in vitro, regulatory gene expression patterns in classical chondrogenesis related markers emerged suggesting that muscle tissue in vitro could if the correct morphogen combinations are applied at the correct temporal intervals direct the tissue to undergo chondrogenesis. rBMP-2 alone was sufficient to initiate a chondrogenesis-like reaction at an early stage. This response was partially inhibited by rTGF-β_3_ and/or rBMP-7 when these were added in conjunction with the rBMP-2. However, whilst it is clear that early stage stimulation is restricted to certain growth factors, growth factor pairs and even all multiple growth factors applied at once are necessary later to modulate and maintain a chondrogenic morphogenesis reaction when using muscle tissue ex vivo. While this new model is still in its early testing stages, requiring further experimental validations in follow up experiments incorporating more gene and protein expression profiles over time, it nonetheless strongly suggests that the correct combination of signalling ligands at the correct time can stimulate and maintain muscle tissue ex vivo to undergo chondrogenesis.

## Figures and Tables

**Figure 1 ijms-21-04863-f001:**
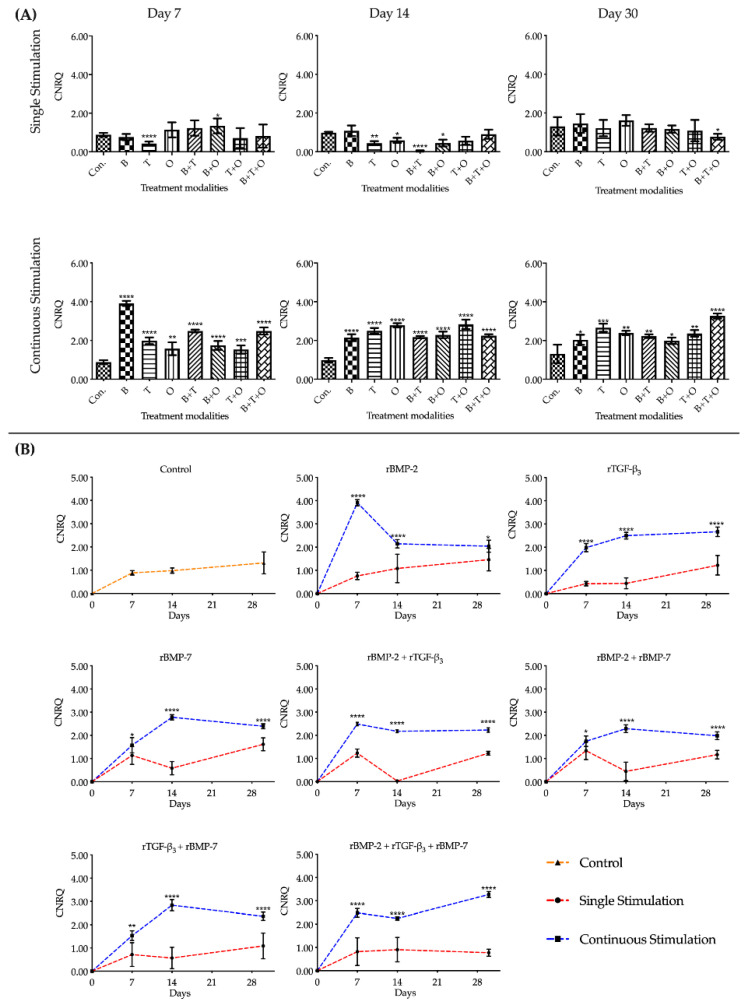
The analyses of the relative gene expression levels of *aggrecan* (*ACAN*). The results were presented as calibrated normalized relative quantity (CNRQ). (**A**) Comparisons between each experimental group under different stimulation modes and the corresponding control group at day 7, 14 and 30 using one-way ANOVA. (**B**) Comparisons between the experimental groups where growth factors were applied continuously and the corresponding 48 h withdrawal groups at day 7, 14 and 30 using two-way ANOVA. Con. = control group, B= rBMP-2 treated group, T = rTGF-β_3_ treated group, O = rBMP-7 treated group, B + T = rBMP-2 + rTGF-β_3_ treated group, B + O = rBMP-2 + rBMP-7 treated group, T + O= rTGF-β_3_ + rBMP-7 treated group, B + T + O = rBMP-2 + rTGF-β_3_ + rBMP-7 treated group. Number of replicates (n) = 6. We defined *p* < 0.05 as a statistically significant difference. * *p* < 0.05, ** *p* < 0.01, *** *p* < 0.001, **** *p* < 0.0001. In (**A**) statistical significance is expressed between each group and the control group (normal medium), while in (**B**) it is expressed in each time point between continuous stimulation and single stimulation. The baseline 0 represents fresh non-cultured muscle tissue which was the normalization factor.

**Figure 2 ijms-21-04863-f002:**
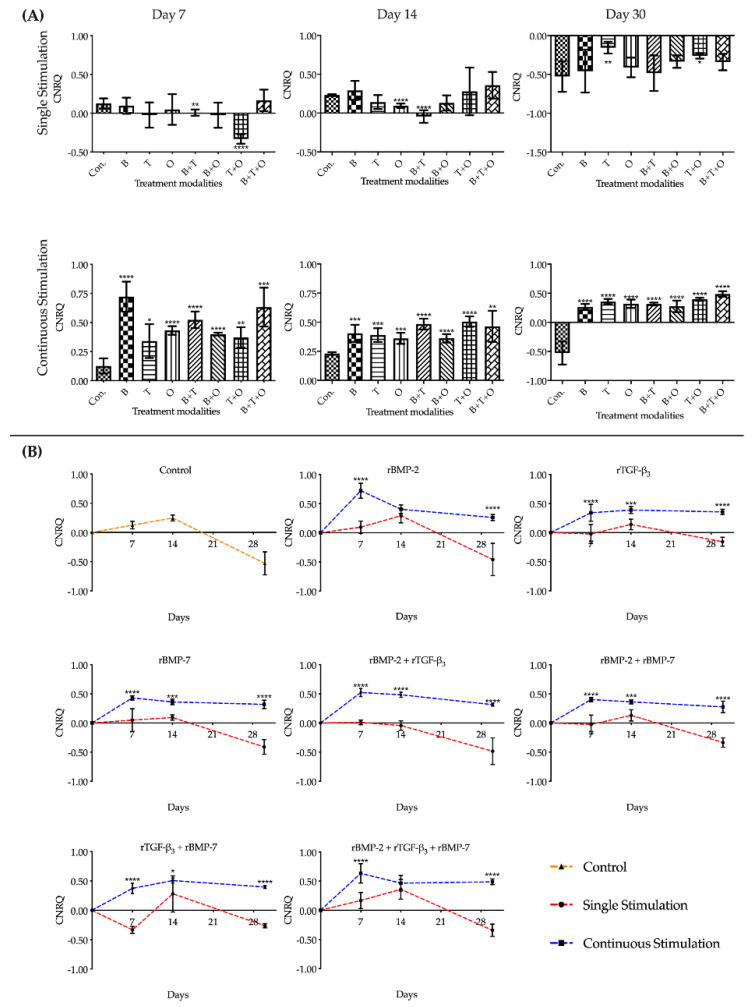
The analyses of the relative gene expression levels of *SRY (Sex Determining Region Y)-Box 9* (*SOX9*). The results were presented as calibrated normalized relative quantity (CNRQ). (**A**) Comparisons between each experimental group under different stimulation modes and the corresponding control group at day 7, 14 and 30 using one-way ANOVA. (**B**) Comparisons between the experimental groups where growth factors were applied continuously and the corresponding 48 h withdrawal groups at day 7, 14 and 30 using two-way ANOVA. Con. = control group, B= rBMP-2 treated group, T = rTGF-β_3_ treated group, O = rBMP-7 treated group, B + T = rBMP-2 + rTGF-β_3_ treated group, B + O = rBMP-2 + rBMP-7 treated group, T + O = rTGF-β_3_ + rBMP-7 treated group, B + T + O = rBMP-2 + rTGF-β_3_ + rBMP-7 treated group. Number of replicates (*n*) = 6. We defined *p* < 0.05 as a statistically significant difference. * *p* < 0.05, ** *p* < 0.01, *** *p* < 0.001, **** *p* < 0.0001. In (**A**) statistical significance is expressed between each group and the control group (normal medium), while in (**B**) it is expressed in each time point between continuous stimulation and single stimulation. The baseline 0 represents fresh non-cultured muscle tissue which was the normalization factor.

**Figure 3 ijms-21-04863-f003:**
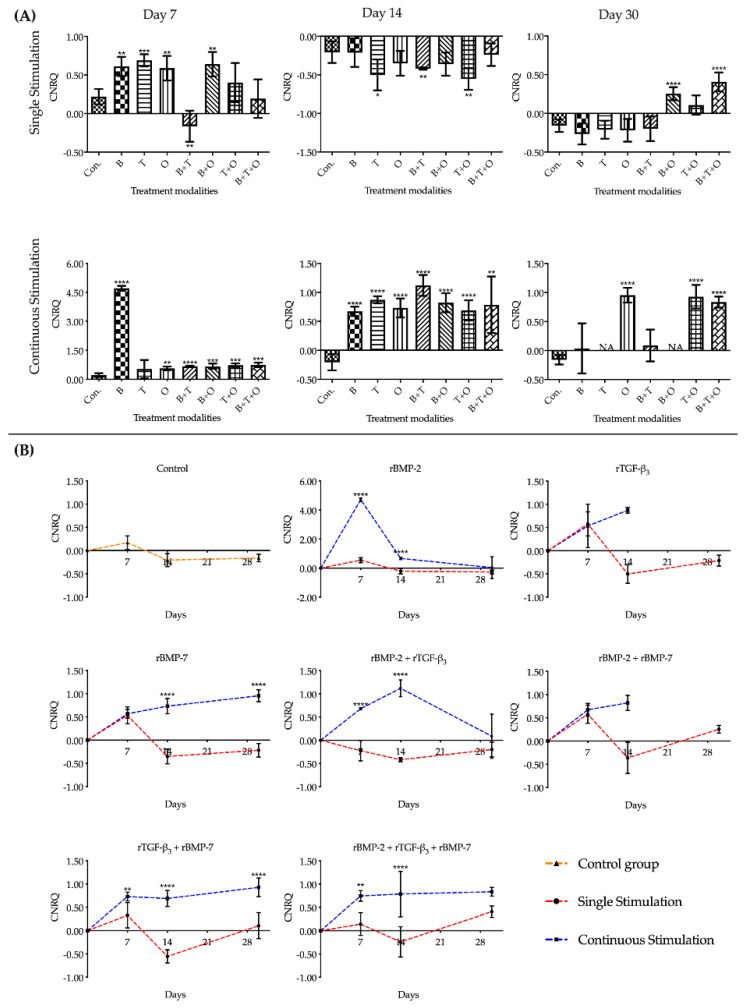
The analyses of the relative gene expression levels of *collagen type II alpha 1* (*Col2α1*). The results were presented as calibrated normalized relative quantity (CNRQ). (**A**) Comparisons between each experimental group under different stimulation modes and the corresponding control group at day 7, 14 and 30 using one-way ANOVA. (**B**) Comparisons between the experimental groups where growth factors were applied continuously and the corresponding 48 h withdrawal groups at day 7, 14 and 30 using two-way ANOVA. The data volume in the experimental groups applied rTGF-β_3_ and rBMP-2 + rBMP-7 continuously at day 30 could not meet the requirements of the statistical analysis. NA = not available. Con. = control group, B = rBMP-2 treated group, T = rTGF-β_3_ treated group, O = rBMP-7 treated group, B + T = rBMP-2 + rTGF-β_3_ treated group, B + O = rBMP-2 + rBMP-7 treated group, T + O = rTGF-β_3_ + rBMP-7 treated group, B + T + O = rBMP-2 + rTGF-β_3_ + rBMP-7 treated group. Number of replicates (*n*) = 6. We defined *p* < 0.05 as a statistically significant difference. * *p* < 0.05, ** *p* < 0.01, *** *p* < 0.001, **** *p* < 0.0001. In (**A**) statistical significance is expressed between each group and the control group (normal medium), while in (**B**) it is expressed in each time point between continuous stimulation and single stimulation. The baseline 0 represents fresh non-cultured muscle tissue which was the normalization factor.

**Figure 4 ijms-21-04863-f004:**
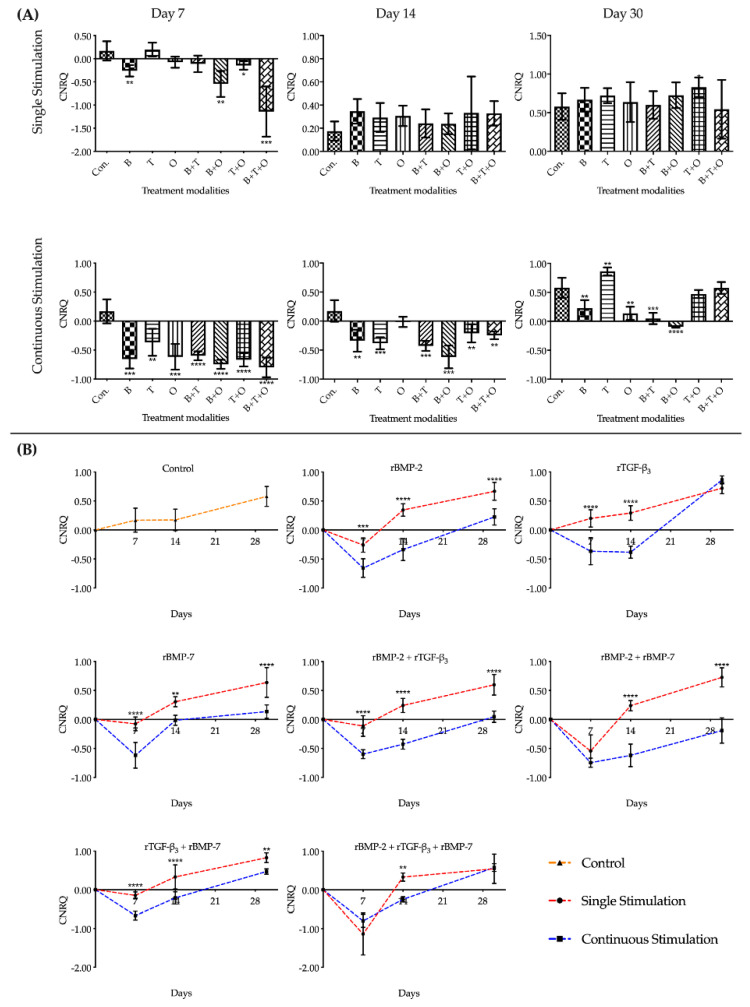
The analyses of the relative expression levels of *collagen type I alpha 1* (*Col1α1*). The results were presented as calibrated normalized relative quantity (CNRQ). (**A**) Comparisons between each experimental group under different stimulation modes and the corresponding control group at day 7, 14 and 30 using one-way ANOVA. (**B**) Comparisons between the experimental groups where growth factors were applied continuously and the corresponding 48 h withdrawal groups at day 7, 14 and 30 using two-way ANOVA. Con. = control group, B= rBMP-2 treated group, T = rTGF-β_3_ treated group, O = rBMP-7 treated group, B + T = rBMP-2 + rTGF-β_3_ treated group, B + O = rBMP-2 + rBMP-7 treated group, T + O = rTGF-β_3_ + rBMP-7 treated group, B + T + O = rBMP-2 + rTGF-β_3_ + rBMP-7 treated group. Number of replicates (*n*) = 6. We defined *p* < 0.05 as a statistically significant difference. * *p* < 0.05, ** *p* < 0.01, *** *p* < 0.001, **** *p* < 0.0001. In (**A**) statistical significance is expressed between each group and the control group (normal medium), while in (**B**) it is expressed in each time point between continuous stimulation and single stimulation. The baseline 0 represents fresh non-cultured muscle tissue which was the normalization factor.

**Figure 5 ijms-21-04863-f005:**
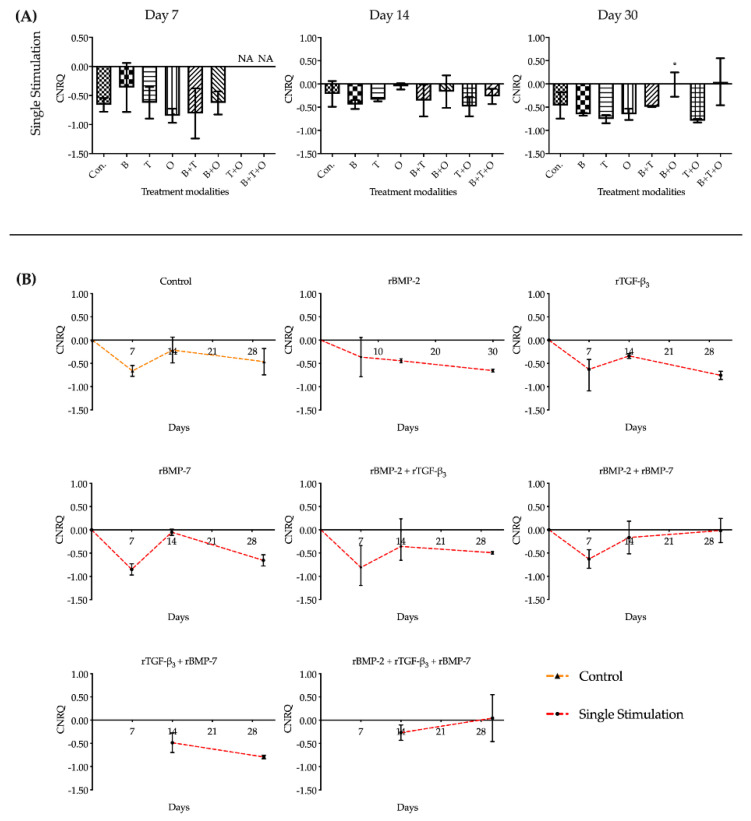
The analyses of the relative expression levels of *collagen type X alpha 1* (*Col10α1*). The results were presented as calibrated normalized relative quantity (CNRQ). (**A**) Comparisons between each experimental group under different stimulation modes and the corresponding control group at day 7, 14 and 30 using one-way ANOVA. (**B**) Comparisons between the experimental groups where growth factors were applied continuously and the corresponding 48 h withdrawal groups at day 7, 14 and 30 using one-way ANOVA. Con.= control group, B= rBMP-2 treated group, T = rTGF-β_3_ treated group, O = rBMP-7 treated group, B + T = rBMP-2 + rTGF-β_3_ treated group, B + O = rBMP-2 + rBMP-7 treated group, T + O = rTGF-β_3_ + rBMP-7 treated group, B + T + O= rBMP-2 + rTGF-β_3_ + rBMP-7 treated group. Number of replicates (*n*) = 6. We defined *p* < 0.05 as a statistically significant difference. * *p* < 0.05. In (**A**) statistical significance is expressed between each group and the control group (normal medium), while in (**B**) it is expressed in each time point between continuous stimulation and single stimulation. The baseline 0 represents fresh non-cultured muscle tissue which was the normalization factor.

**Figure 6 ijms-21-04863-f006:**
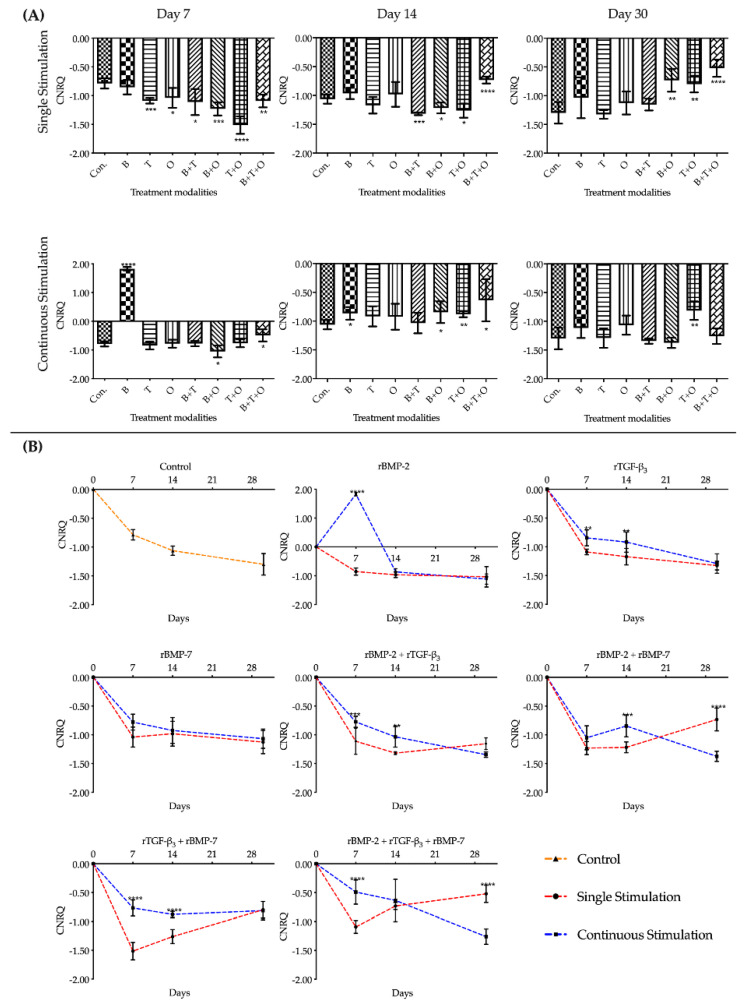
The analyses of the relative expression levels of *alkaline phosphatase* (*ALP*). The results were presented as calibrated normalized relative quantity (CNRQ). (**A**) Comparisons between each experimental group under different stimulation modes and the corresponding control group at day 7, 14 and 30 using one-way ANOVA. (**B**) Comparisons between the experimental groups where growth factors were applied continuously and the corresponding 48 h withdrawal groups at day 7, 14 and 30 using two way ANOVA. Con. = control group, B = rBMP-2 treated group, T = rTGF-β_3_ treated group, O = rBMP-7 treated group, B + T = rBMP-2 + rTGF-β_3_ treated group, B + O = rBMP-2 + rBMP-7 treated group, T + O = rTGF-β_3_ + rBMP-7 treated group, B + T + O = rBMP-2 + rTGF-β_3_ + rBMP-7 treated group. Number of replicates (*n*) = 6. We defined *p* < 0.05 as a statistically significant difference. * *p* < 0.05, ** *p* < 0.01, *** *p* < 0.001, **** *p* < 0.0001. In (**A**) statistical significance is expressed between each group and the control group (normal medium), while in (**B**) it is expressed in each time point between continuous stimulation and single stimulation. The baseline 0 represents fresh non-cultured muscle tissue which was the normalization factor.

**Figure 7 ijms-21-04863-f007:**
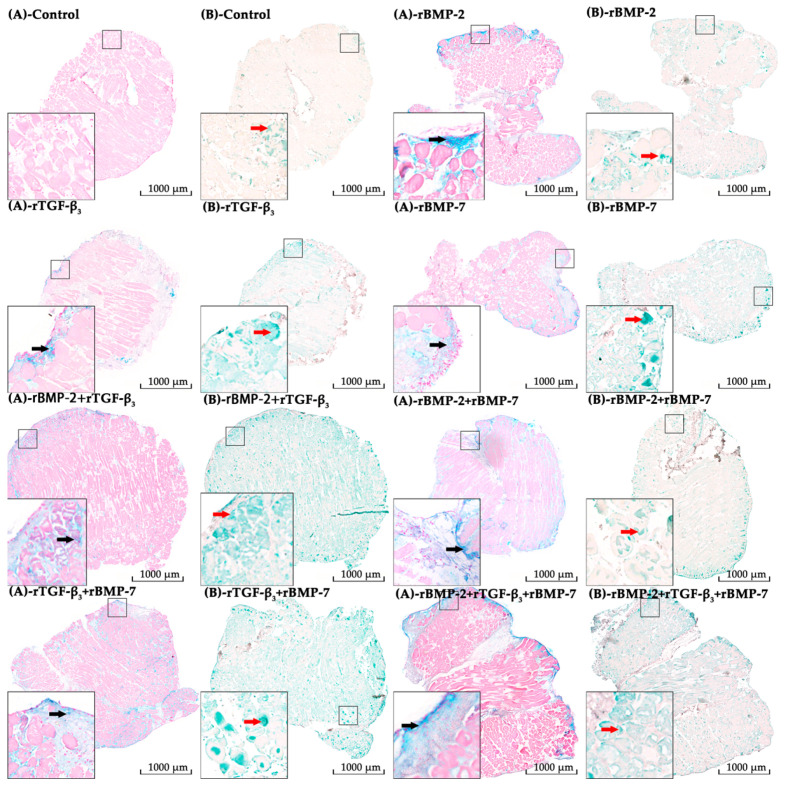
The staining results of the control group and the experimental groups stimulated continuously harvested at culture day 30, including alcian blue staining and immunohistochemistry. (**As**) Alcian blue staining and (**Bs**) immunohistochemistry aggrecan results of the control group and the experimental groups in which morphogens were applied continuously. Positive alcian blue indicated the deposition of low acidic/carboxylated polysaccharides (**black arrows**) with positive antigen-antibody interactions in immunohistochemistry being green (**red arrow**). The magnification was set at 40×.

**Figure 8 ijms-21-04863-f008:**
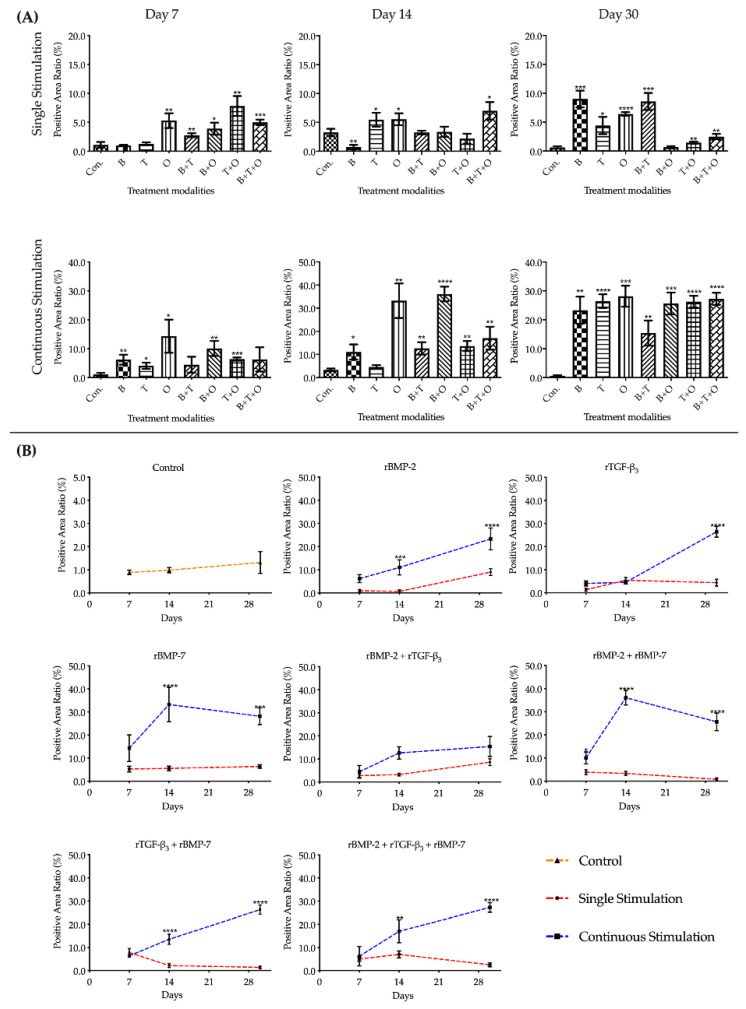
The semi-quantitative histomorphometrical analysis of alcian blue staining. The results were presented as positive area ratio (%). (**A**) Comparisons between each experimental group under different stimulation modes and the corresponding control group at day 7, 14 and 30 using one-way ANOVA. (**B**) Comparisons between the experimental groups where growth factors were applied continuously and the corresponding 48 h withdrawal groups at day 7, 14 and 30 using two-way ANOVA. Con. = control group, B= rBMP-2 treated group, T= rTGF-β_3_ treated group, O = rBMP-7 treated group, B + T = rBMP-2 + rTGF-β_3_ treated group, B + O = rBMP-2 + rBMP-7 treated group, T + O= rTGF-β_3_ + rBMP-7 treated group, B + T + O= rBMP-2 + rTGF-β_3_ + rBMP-7 treated group. Number of replicates (*n*) = 6. We defined *p* < 0.05 as a statistically significant difference. * *p* < 0.05, ** *p* < 0.01, *** *p* < 0.001, **** *p* < 0.0001. In (**A**) statistical significance is expressed between each group and the control group (normal medium), while in (**B**) it is expressed in each time point between continuous stimulation and single stimulation.

**Figure 9 ijms-21-04863-f009:**
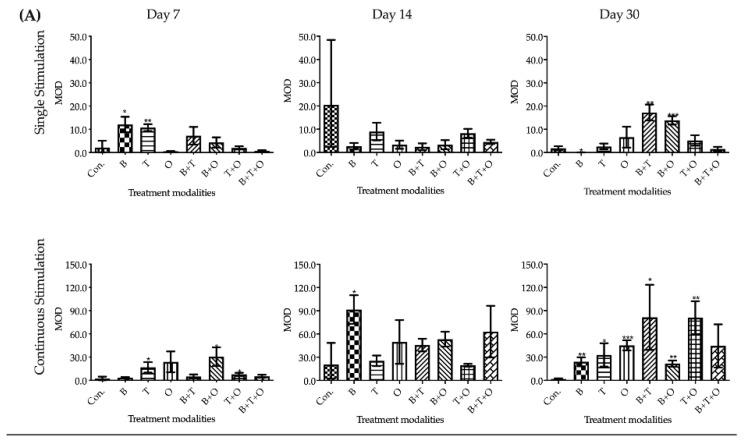
The semi-quantitative analysis results of immunohistochemistry. The results were presented as mean optical density value (MOD), which reflect the density of the antigen. The target antigen in immunohistochemistry was aggrecan. (**A**) Comparisons between each experimental group under different stimulation modes and the corresponding control group at day 7, 14 and 30 using one-way ANOVA. (**B**) Comparisons between the experimental groups where growth factors were applied continuously and the corresponding 48 h withdrawal groups at day 7, 14 and 30 using two-way ANOVA. Con. = control group, B = rBMP-2 treated group, T = rTGF-β_3_ treated group, O = rBMP-7 treated group, B + T = rBMP-2 + rTGF-β_3_ treated group, B + O = rBMP-2 + rBMP-7 treated group, T + O = rTGF-β_3_ + rBMP-7 treated group, B + T + O = rBMP-2 + rTGF-β_3_ + rBMP-7 treated group. Number of replicates (*n*) = 6. We defined *p* < 0.05 as a statistically significant difference. * *p* < 0.05, ** *p* < 0.01, *** *p* < 0.001, **** *p* < 0.0001. In (**A**) statistical significance is expressed between each group and the control group (normal medium), while in (**B**) it is expressed in each time point between continuous stimulation and single stimulation.

**Figure 10 ijms-21-04863-f010:**
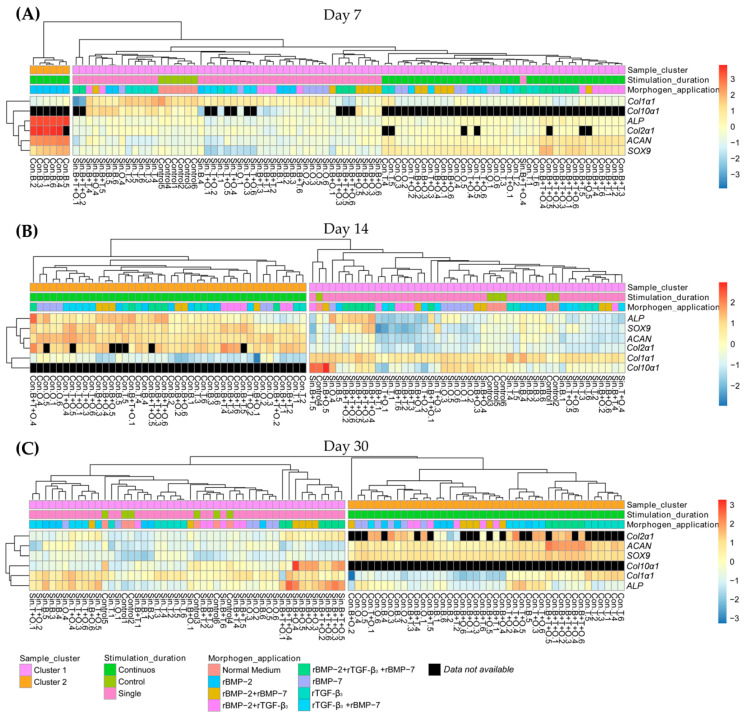
Hierarchical clustering of gene expression of all the samples at day 7 (**A**), 14 (**B**) and 30 (**C**). The distance measure used in the clustering rows and column was Euclidean, the agglomeration method used to cluster was the average. Con. = Continuous, Sin. = Single, B= rBMP-2 treated group, T = rTGF-β_3_ treated group, O = rBMP-7 treated group, B + T = rBMP-2 + rTGF-β_3_ treated group, B + O = rBMP-2 + rBMP-7 treated group, T + O = rTGF-β_3_ + rBMP-7 treated group, B + T + O = rBMP-2 + rTGF-β_3_ + rBMP-7 treated group, *ACAN = aggrecan*, *SOX9 = sex-determining region Y (SRY)-box 9*, *Col2α1 = collagen type II alpha 1*, *Col1α1 = collagen type I alpha 1*, *Col10α1* = *collagen type X alpha 1, ALP = alkaline phosphatase.*

**Figure 11 ijms-21-04863-f011:**
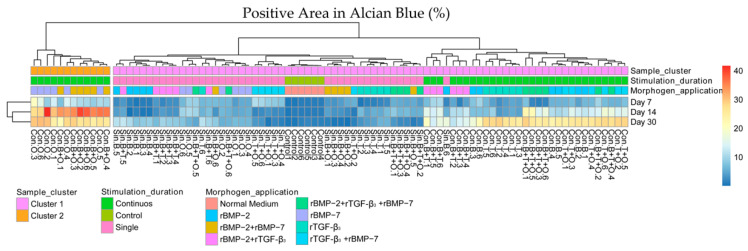
Hierarchical clustering of the alcian blue staining area of all the samples at all culturing time points. The distance measure used in the clustering rows and column was Euclidean, the agglomeration method used to cluster was the average. Con. = Continuous, Sin. = Single, B = rBMP-2 treated group, T = rTGF-β_3_ treated group, O = rBMP-7 treated group, B + T = rBMP-2 + rTGF-β_3_ treated group, B + O = rBMP-2 + rBMP-7 treated group, T + O = rTGF-β_3_ + rBMP-7 treated group, B + T + O = rBMP-2 + rTGF-β_3_ + rBMP-7 treated group.

**Table 1 ijms-21-04863-t001:** Gene primers for Rattus norvegicus with accession number and sequence.

	Gene	Accession Number	5′ Sequence	3′ Sequence
Reference genes	*TBP*	BC081939.1	TAACCCAGAAAGTCGAAGAC	CCGTAAGGCATCATTGGA
	*GAPDH*	BC083511.1	CATGGGTGTGAACCATGA	TGTCATGGATGACCTTGG
	*POLR2e*	BC158787.1	GACCATCAAGGTGTACTGC	CAGCTCCTGCTGTAGAAAC
	*RPLP0*	BC001834.2	CAACCCAGCTCTGGAGA	CAGCTGGCACCTTATTGG
	*SDHA*	NM_130428.1	GCGGTATGAGACCAGTTATT	CCTGGCAAGGTAAACCAG
	*RPL13α*	NM_173340.2	TTTCTCCGAAAGCGGATG	AGGGATCCCATCCAACA
Genes of interest	*Col1α1*	NM_053304.1	GGTGACAGAGGCATAAAGG	AGACCGTTGAGTCCATCT
	*Col2α1*	NM_012929.1	ATCCAGGGCTCCAATGA	TCTTCTGGAGTGCGGAA
	*Col10α1*	XM_001053056.7	CCAGGTCTCAATGGTCCTA	ATTTCCTCACGGACCTGT
	*ACAN*	NM_022190.1	CAAGTGGAGCCGTGTTT	TTTAGGTCTTGGAAGCGAG
	*ALP*	NM_013059.2	CGACAGCAAGCCCAAG	AGACGCCCATACCATCT
	*SOX9*	NM_080403.1	CCAGAGAACGCACATCAAG	ATACTGATGTGGCTGGTGG

TBP: TATA-binding protein, GAPDH: Glyceraldehyde 3-phosphate dehydrogenase, POLR2e: RNA polymerase II subunit e, RPLP0: Ribosomal protein lateral stalk subunit P0, SDHA: Succinate dehydrogenase complex flavoprotein sub-unit A, RPL13α: Ribosomal protein L13α, Col1α1: Collagen Type I Alpha 1, Col2α1: Collagen Type II Alpha 1, Col10α1: Collagen Type X Alpha 1, ACAN: Aggrecan, ALP: Alkaline phosphates, SOX9: SRY (Sex Determining Region Y)-Box 9.

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
