# Peer review of "Temporal TGF-β Supergene Family Signalling Cues Modulating Tissue Morphogenesis: Chondrogenesis within a Muscle Tissue Model?"

_ijms, 2020, doi:10.3390/ijms21144863_

Round 1
Reviewer 1 Report
In the present manuscript with reference IJMS-815627 and entitled: “Temporal TGF-β supergene family signalling cues modulating tissue morphogenesis: chondrogenesis within a muscle tissue model?”, authors explore the effects of three different morphogens (BMP2, TGFB and BMP7) and with different combinations (in time and in morphogens) of treatment over a muscle tissue in order to demonstrate that a particular modulation of this tissue might be possible, and particularly towards a chondrogenesis. For such goal authors explore the gene expression of different genes known to be involved in endochondral bone formation (Sox9, Aggrecan, Col1a1, Clo2a1, Col10a1 and ALP) at 7, 14 and 30 days post-treatment with morphogens as well as through histomorphological (alcian blue) and immunohistochemistry (aggrecan) analyses. The study is interesting within tissue engineering and regenerative medicine research lines.
Overall, the manuscript clearly requires some modifications on the text, and how results are shown in order to improve the clarity and to better illustrate how the different combinations of morphogens might have a different impact on the muscle. In addition, the research work could gain more robustness if authors perform some additional analysis. These and another small comments/suggestions are detailed as follows:
- Cited literature is somehow “old”. Most of the articles cited regarding what has been done in tissue engineering and regenerative medicine in articular cartilage are from 80 to 200, and only very few were from 2010-2020 regardless of the enormous advance on this research lines during the last decade. Particular examples of outdated literature are references 6, 17-18, 20, 23, 26, 27, etc…. For instance, Komori 2010 provide a clear picture of the role of Runx2 in osteo- and chondro-genesis regulation
- Lines 46-47: …. among another biological processes.
- Lines 51-52: I suggest authors to use the term BMP7 over OP-1 since the first is more commonly used as well as it is recommended by Uniprot database.
- Lines 69-75: how some cell types are known to be able to trans-differentiate and/or de-differentiate. Manuscript might be benefited by authors highlighting the additional value of differentiated cells (if they had one) versus the use of stem cell population. What has been done with stem cells so far? Please, also provide positive and negative aspect of both approaches. Some text in this regard has been found in the discussion section….
- In the introduction, authors should provide a brief description of the genes implicated in osteogenesis and chondrogenesis, mostly those here analyzed in order to not familiar readers get an idea why those are analyzed.
- One of the major concerns to consider that this manuscript needs major revision is that the way of presenting data is quite difficult to compare, dissect, interpret and discus. In addition, sometimes data presented is quite redundant. Please, let me describe my impression in this regard in more detail. For each gene, authors presented twice the results of qPCR. One comparing the data withing each sampling time (histograms) and after comparing withing each treatment (morphogen exposure) along time. Furthermore, data and statistics are presented in additional data. I understand that in this way comparisons are easier to perform, but It might be better to reduce a little bit the redundancy. What about organizing them in another way. For instance, since at each sampling point differences are indicated with asterisks, differences among sampling point can be indicated with letters (which is more appropriated in this case). Thus, author can present both analysis in the same picture (with a bigger size). Moreover, I strongly suggest authors to perform a hierarchical clustering with all the data (gene expression, histomorphology and immunohistochemistry) in order to compare and “cluster” the experimental treatments which might help to identify which treatments are profiling the chondrogenic process in the same manner.
- Another major issue, and perhaps an additional analysis to be performed is the gene expression of more interesting genes to further characterize the chondrogenetic process here hypothesized. In this sense, Run2 might be a clear example of an important osteo/chondrogenesis regulator. Moreover, why authors do not explored the expression of more specific articular cartilage markers? For instance, ABI3BP, THBS4 and SIX1 (from Hissnauer et al., 2010; https://doi.org/10.1016/j.joca.2010.10.002)
- Lines 95-103: this is materials and methods.
- Lines 115-117: First sentence is abruptly ended?
- Results: authors stated that a Two-way ANOVA was performed, and in figures and supplementary data the statistical differences are shown. However, there is no indication if there is a significant interaction between the two factors considered (time and morphogen). Please, provide whether this interaction occurs.
- Figures: please, indicate what is represented in the y axis. Second increase the size of name of the x axis. Add in the figure legend when an one-way or two-way ANOVA was performed and the number of replicates.
- Regarding increased or decreased gene expression. Have authors explored how the morphogens control cell proliferation? Which different cell types are present and how they proliferate might underlie why upon different morphogens the tissue respond in one or another way.
- In some genes, authors could not detect their expression but they do afterwards. Although authors try to explain this, I guess they should make a bigger effort to explain it in a more convincing way.
- Authors perform a relative gene expression analysis. But which sample and sampling time was selected as the reference (and set to 1)?
- Line 207: minuscule? Please, be objective and try to not use this kind of adjectives. Is the downregulation significant or not?
- Line 220: minimally,…. same comment as before.
- Lines 250-253: in control the area was lower to 5%. What is the alcian blue stained area represented in all the experimental treatments?
- Legend figure 4: please, revise it since it is stated as in both cases the morphogens were applied continuously? Also, no green color is appreciated in the figure,.... authors should present a magnification of the image?
- Figure 6: Some strange data is presented. First, single application, at day 14 the immunohistochemistry histogram of the control group is bigger than the other treatments, it seems there is a big variability on this parameter among the experimental groups and the sampling time. Second, continuous application, day 14, B+T although haven a low variation is not significant different from control? A clear image of the immunostaining should be presented.
- Discussion, there is not clear the biological process to which the muscle tissue is driven. Authors talk about chondrogenic process but no endochondral, correlating with this the lack of gene expression of Col2a1 and col10a1 (the last marker of hypertrophic chondrocytes). Some speculation has been presented towards an articular chondrogenesis, but no specific marker of articular cartilage has been evaluated to sustain this (please see previous comment on this regard; Hissnauer et al., 2010). In lines 316-320, authors should interpret the results base on what is presented and the last hypothesis is not sustained by reported data.
- Lines 349-354: how authors are sure that the morphogens are only acting on the already differentiated myocytes and not on the stem cell population included in the tissue?
- Lines 379-382: as previously commented, a hierarchical clustering of the different experimental groups based on the gene expression and alcian blue and aggrecan staining might help to visualize the effects and outcomes of the different combinations of morphogens.
- Lines 391-395: authors could explore with SMAD isoforms were phosphorylated in order to get further insights on this issue....
- Line 403: as commented by the authors the present study is limited by only exploring one specific concentration of the morphogens.
- Authors should be cautious. Authors only explore its expression at 3 specific time points, not all the process during the 30 days. In addition, hypertrophic chondrocytes may undergo apoptosis afterwards, and thus, this may explain why at particular sampling points there is no col10 gene expression....
- Lines 411-414: please, rephrase as it is difficult to identify the goal of this sentences.....
- Lines 415-416: the fibro-chondrogenesis is not only characterized y Col1a1, nor the col1a1 is a specific marker of fibro-chondrogenesis. Please, re-phrase.
- Line 420: 90-95
- Lines 423-424: again, since authors did not explore the specific markers of articular cartilage, I would say chondrogenesis, and not articular chondrogenesis, please see article Hissnauer et al., 2010.
- Lines 433-435: If authors talk about fibro-chondrogenesis, it seems clear that this process here described has been identified and characterized by other researchers and pathologists before. Please, re-phrase it accordingly.
- Lines 468-470: it is unclear whether author only evaluate the gene expression of reference genes in fresh muscle, or in all sampling times (7, 14 and 30 days). Please specify it.
- Lines 533-534: please, indicate the values of the 260/280 and 230/260 ratios to show the purity of the mRNA samples after nucleic acid isolation.
- Line 560: please, indicate the pH at which the alcian blue has been done since the type of proteoglycans stained might be different depending the ph.
- Conclusions: base on the lack of specific markers for articular chondrogenesis, present results suggest a kind of chondrogenesis, but it could be osteo- and/or fibro-chondrogenesis. Authors base on the presented data cannot conclude if this is fibro- or articular-chondrogenesis.
- Line 607: authors only used one specific type of tissue, muscle. Please, re-phrase it.
- Conclusions are a little bit vague. Can authors conclude something more specific, for instance which the combination used of morphogens were the most successful to induce a chondrogenesis process?
Author Response
Overall response to reviewers’ comments
In the present manuscript with reference IJMS-815627 and entitled: “Temporal TGF-β supergene family signalling cues modulating tissue morphogenesis: chondrogenesis within a muscle tissue model?”, authors explore the effects of three different morphogens (BMP2, TGFB and BMP7) and with different combinations (in time and in morphogens) of treatment over a muscle tissue in order to demonstrate that a particular modulation of this tissue might be possible, and particularly towards a chondrogenesis. For such goal authors explore the gene expression of different genes known to be involved in endochondral bone formation (Sox9, Aggrecan, Col1a1, Clo2a1, Col10a1 and ALP) at 7, 14 and 30 days post-treatment with morphogens as well as through histomorphological (alcian blue) and immunohistochemistry (aggrecan) analyses. The study is interesting within tissue engineering and regenerative medicine research lines.
Overall, the manuscript clearly requires some modifications on the text, and how results are shown in order to improve the clarity and to better illustrate how the different combinations of morphogens might have a different impact on the muscle. In addition, the research work could gain more robustness if authors perform some additional analysis. These and another small comments/suggestions are detailed as follows.
Response:
We would like to thank the reviewer for his/her appraisal of the work and thank him/her for pointing out critical aspects to make this manuscript suitable for publication within the Journal. Most of all the comments have been addressed that the reviewer has suggested including improving the quality of the Scientific English of the manuscript.
Point by point response
- Cited literature is somehow “old”. Most of the articles cited regarding what has been done in tissue engineering and regenerative medicine in articular cartilage are from 80 to 200, and only very few were from 2010-2020 regardless of the enormous advance on this research lines during the last decade. Particular examples of outdated literature are references 6, 17-18, 20, 23, 26, 27, etc…. For instance, Komori 2010 provide a clear picture of the role of Runx2 in osteo- and chondro-genesis regulation
Response: We thank the reviewer for pointing this out. We have gone through the manuscript and where relevant updated some of the literature to better reflect the context of discoveries over the last decade. However, some of the references mentioned by the reviewer are key fundamental discoveries that are relevant and have to be included. We have however cut down where relevant to accommodate the reviewers request and indicated reference changes in yellow.
- Lines 46-47: …. among another biological processes.
Response: We thank the reviewer for this comment. We did not quite understand what the wishes of the reviewer where in this regard. We interpreted it that the reviewer wanted this added to talk about this in the introduction with regard to the previous sentence. We have tried to accommodate this (Page 2 Line 45).
- Lines 51-52: I suggest authors to use the term BMP7 over OP-1 since the first is more commonly used as well as it is recommended by Uniprot database.
Response: We thank the reviewer for pointing this out. We have made the relevant alteration but do in the first instance also point towards the alternative name.
- Lines 69-75: how some cell types are known to be able to trans-differentiate and/or de-differentiate. Manuscript might be benefited by authors highlighting the additional value of differentiated cells (if they had one) versus the use of stem cell population. What has been done with stem cells so far? Please, also provide positive and negative aspect of both approaches. Some text in this regard has been found in the discussion section….
Response: We thank the reviewer for pointing this out. We have briefly expanded the introduction where applicable (page 2 line 74-77).
- In the introduction, authors should provide a brief description of the genes implicated in osteogenesis and chondrogenesis, mostly those here analyzed in order to not familiar readers get an idea why those are analyzed.
Response: We thank the reviewer for pointing this out. We have briefly highlighted important gene markers in the osteogenesis and chondrogenesis related pathways and why the subset of genes was chosen for the work (page 2-3 lines 91-97).
- One of the major concerns to consider that this manuscript needs major revision is that the way of presenting data is quite difficult to compare, dissect, interpret and discus. In addition, sometimes data presented is quite redundant. Please, let me describe my impression in this regard in more detail. For each gene, authors presented twice the results of qPCR. One comparing the data withing each sampling time (histograms) and after comparing withing each treatment (morphogen exposure) along time. Furthermore, data and statistics are presented in additional data. I understand that in this way comparisons are easier to perform, but It might be better to reduce a little bit the redundancy. What about organizing them in another way. For instance, since at each sampling point differences are indicated with asterisks, differences among sampling point can be indicated with letters (which is more appropriated in this case). Thus, author can present both analysis in the same picture (with a bigger size). Moreover, I strongly suggest authors to perform a hierarchical clustering with all the data (gene expression, histomorphology and immunohistochemistry) in order to compare and “cluster” the experimental treatments which might help to identify which treatments are profiling the chondrogenic process in the same manner.
Response: We thank the reviewer for his/her suggestion in how the figures could be improved. We have consider various possibilities on how to best present the figures, and we do agree that the reviewers comments do help us to provide a more intuitive and clear picture to readers when interpreting the results, thereby possibly avoiding data redundancy. Whilst we have discussed the possibility, the figures as we have present, in our opinion, fully express our comparison results for the three independent factors that we discuss in our study. Our reasons are as follows:
For example, taking Figure-1, part (A) focused on the comparison between different experimental groups to control groups (shown by asterisks), in which the first row is showing only ligand application for 48h. The second row is the continuous stimulation, where the first column to the third presents the different harvesting times (day 7, 14, 30). Part B contains two independent variables, namely, different stimulation duration (48h and continuously) with regulation of gene expression over time (7, 14, 30). The eight small figures correspond to the eight morphogen treatment groups (Tissue+normal medium, + BMP-2, + TGF- β3 , + BMP-7, etc…). The asterisks in part B of the figure represents the difference between 48h stimulation vs continuous stimulation in the same morphogen modality at a certain sampling time, while the relatively less important results that showed the changes over time were presented in the form of tables in the supplementary materials.
In relation to the reviewer's opinion, "…for instance, since at each sampling point differences are indicated with asterisks, differences among sampling point can be indicated with letters (which is more approved in this case), Author can present both analysis in the same picture (with a bigger size). ", whilst we can combine the data of each sampling time (histograms) and each treatment in one figure, we would lose the effect of comparing different stimulation durations (48h vs continuously) for the gene expression. Although some of the data are redundant, it is also an insurmountable situation due to too many independent factors that we explored.
Moreover, we have also carefully considered "a hierarchical clustering with all the data" proposed by the reviewer. For example, we tried to cluster the results according to different morphogen modality (Tissue+normal medium, + BMP-2, + TGF- β3, + BMP-7, etc…) and combine the gene data with staining and immunohistochemistry data, but again we would lose the comparison between the different modalities, whilst at the same time still causing data redundancy. The same problem also arose when we clustering the results according to the stimulation duration (48h vs continuously) and sampling time (Day 7, 14, 30).
As such whichever way we try it our representation of the data is still the most logical but we will see how we can alleviate this issue in the future when dealing with massive amounts of data that such experiments inevitably cause.
- Another major issue, and perhaps an additional analysis to be performed is the gene expression of more interesting genes to further characterize the chondrogenetic process here hypothesized. In this sense, Run2 might be a clear example of an important osteo/chondrogenesis regulator. Moreover, why authors do not explored the expression of more specific articular cartilage markers? For instance, ABI3BP, THBS4 and SIX1 (from Hissnauer et al., 2010; https://doi.org/10.1016/j.joca.2010.10.002)
Response: We thank the reviewer for having made us aware of this criterion. Whilst Hissnauer is correct in suggesting that these additional markers are important for articular chondrogenesis the classical markers, as we have chosen, remain the foundation that over decades of research have remained consistently accurate to help distinguish chondrogenesis pathways and what is being formed. The additional markers we intend to incorporate in more complex follow-up studies around the model system we have presented here in this manuscript that will in greater detail also include osteogenesis related pathways. Follow-up studies will expand on this greatly.
- Lines 95-103: this is materials and methods.
Response: We thank the reviewer for pointing this out. This section has been removed accordingly.
- Lines 115-117: First sentence is abruptly ended?
Response: We thank the reviewer for pointing this out. The manuscript has been thoroughly checked to ensure proper flow of the material.
- Results: authors stated that a Two-way ANOVA was performed, and in figures and supplementary data the statistical differences are shown. However, there is no indication if there is a significant interaction between the two factors considered (time and morphogen). Please, provide whether this interaction occurs.
Response: We thank the reviewer for pointing this out. The interactions between the stimulation duration (only 48h or continuous) and the culture sampling time (7, 14 or 30 days) in most treatment modalities groups were significant when the gene expression and the histomorphometrical analysis were analysed using 2 way ANOVA. Where relevant we have now included this in the results to make it clearer (Page 3 line 118-120; page 5 line 170-171; page 6 line 195-199; page 6 line 219-221; page 9 line 270-271; page 10 line 289-292; page 12 line 331-334; page 14 line 376-380.).
The special situation of different analysis have been explained in the results of each part. The interaction between the stimulation duration and the culture sampling time in the rBMP-7 treated group was not significant when analysing the expression of ALP and the main effects of these two factors were also not significant. The interaction was also not significant in the rBMP-2 + rTGF-β3 treated group when analysed the alcian blue staining, while the main effects of these two factors were significant. The interactions were not significant except in the rTGF-β3, rBMP-7 and rBMP-2 + rTGF-β3 + rBMP-7 treated groups when analysing the aggrecan immunohistochemistry and the main effects of these two factors were significant in rBMP-2 + rTGF-β3 + rBMP-7 treated group, while the main effect was only significant of stimulation duration in rTGF-β3 and rBMP-7 groups. It should be noted that to avoid statistical errors and facilitate the accuracy of interpretation, we removed all the asterisks in the figures, once the interaction was not significant, although the difference of the main effects may be significant.
- Figures: please, indicate what is represented in the y axis. Second increase the size of name of the x axis. Add in the figure legend when an one-way or two-way ANOVA was performed and the number of replicates.
Response: We thank the reviewer for pointing this out. The Figures have been adjusted accordingly and the Legends/Captions now reflect the relevant material. Also we have indicated what the baseline/normalisation factor was to which samples normalised to in the gene expression data including the number of replicates (n=6).
- Regarding increased or decreased gene expression. Have authors explored how the morphogens control cell proliferation? Which different cell types are present and how they proliferate might underlie why upon different morphogens the tissue respond in one or another way.
Response: We thank the reviewer for pointing this out. We direct the reviewer to the response under point 7 where this is currently being investigated in follow up experiments.
- In some genes, authors could not detect their expression but they do afterwards. Although authors try to explain this, I guess they should make a bigger effort to explain it in a more convincing way.
Response: We thank the reviewer for pointing this out. We have included a statement regarding this why this may have happened in the discussion (page 17 line 510-512)
- Authors perform a relative gene expression analysis. But which sample and sampling time was selected as the reference (and set to 1)?
Response: We thank the reviewer for pointing this out. We have indicated this clearly in the materials and methods section but have, to make it clearer, incorporated a brief sentence in each Figure legend where applicable what the normalisation material was. Also since the data is presented as CNRQ (or better log102ΔΔCq or logNRQ) and not fold difference the baseline is 0. The reviewer is directed to reading the combined works of Bustin et al 2009, 2011 and 13 for the advanced qPCR representations of gene data. Fold differences and baselines being 1 are no longer the standards for publishing gene related data with CNRQ properly showing up-or downregulation of gene where applicable.
- Line 207: minuscule? Please, be objective and try to not use this kind of adjectives. Is the downregulation significant or not?
- Line 220: minimally,…. same comment as before.
Response: We thank the reviewer for pointing this out. All superficial wording has been removed.
- Lines 250-253: in control the area was lower to 5%. What is the alcian blue stained area represented in all the experimental treatments?
Response: We thank the reviewer for pointing this out. We have indicated by means of a range what this means (page 12 line 326-327).
- Legend figure 4: please, revise it since it is stated as in both cases the morphogens were applied continuously? Also, no green color is appreciated in the figure,... authors should present a magnification of the image?
Response: We thank the reviewer for pointing this out. Based also on the comments by the second reviewer we have adjusted the Figure slightly to better reflect what is meant by this and the Legend has been tweaked accordingly. (page 11 line 314-322). To clarify, we only presented the figure of control and all the continuous stimulation experimental groups harvested at day 30. We did not present the figures of single stimulation groups or at other time points, because the most distinctive changes were seen at day 30.
- Figure 6: Some strange data is presented. First, single application, at day 14 the immunohistochemistry histogram of the control group is bigger than the other treatments, it seems there is a big variability on this parameter among the experimental groups and the sampling time. Second, continuous application, day 14, B+T although haven a low variation is not significant different from control? A clear image of the immunostaining should be presented.
Response: We thank the reviewer for pointing this out. Although we tried to decrease the variation by excluding the outliers, the variation was still larger than the other groups, hence, we presented the original data. However, when assessing the figures again we noticed that the analysis software we had used to generate the figures in part (A), the mean ± SEM was shown and not the mean ± SD as we had originally intended. As such we have adjusted the error bars of all figures in part (A) to properly reflect this which did not affect the accuracy of the hypothesis tested. Thank you again for making us aware of this error.
- Discussion, there is not clear the biological process to which the muscle tissue is driven. Authors talk about chondrogenic process but no endochondral, correlating with this the lack of gene expression of Col2a1 and col10a1 (the last marker of hypertrophic chondrocytes). Some speculation has been presented towards an articular chondrogenesis, but no specific marker of articular cartilage has been evaluated to sustain this (please see previous comment on this regard; Hissnauer et al., 2010). In lines 316-320, authors should interpret the results base on what is presented and the last hypothesis is not sustained by reported data.
Response: We thank the reviewer for pointing this out. However, classical markers to distinguish between the various chondrogenic process are ACAN, Sox9, Col2, Col1 and ColX specifically. If the relevant matrix protein is not being expressed/synthesised then a specific type of chondrogenesis pathway is being followed. This remains a fundamental fact. However, to meet with the request by the reviewer we have adjusted the section to more appropriately reflect the reviewers wishes (page 14-15 line 396-410).
- Lines 349-354: how authors are sure that the morphogens are only acting on the already differentiated myocytes and not on the stem cell population included in the tissue?
Response: We thank the reviewer for pointing this out. We have not considered this aspect and have discussed it where relevant (page 15 line 443-455).
- Lines 379-382: as previously commented, a hierarchical clustering of the different experimental groups based on the gene expression and alcian blue and aggrecan staining might help to visualize the effects and outcomes of the different combinations of morphogens.
Response: We thank the reviewer for pointing this out. We refer the Reviewer to the previous response under comment 6 that deals with why we chose this figure configuration.
- Lines 391-395: authors could explore with SMAD isoforms were phosphorylated in order to get further insights on this issue....
- Line 403: as commented by the authors the present study is limited by only exploring one specific concentration of the morphogens.
Response: We thank the reviewer for pointing this out. We will investigate this in follow up experiments.
- Authors should be cautious. Authors only explore its expression at 3 specific time points, not all the process during the 30 days. In addition, hypertrophic chondrocytes may undergo apoptosis afterwards, and thus, this may explain why at particular sampling points there is no col10 gene expression....
Response: We thank the reviewer for pointing this out. The figures as such have dotted lines running between the time points as we do not know what the gene expression pattern is like between day 0-7, 7-14 and 14-30. Only detailed analysis can generate this but based on the results a certain theoretical pattern can be deduced. The text also reflects this more adequate. As for the latter comment we have incorporated this reasoning for not being able to detect certain genes at certain time points in conjunction with what we have expanded upon under comment 13.
- Lines 411-414: please, rephrase as it is difficult to identify the goal of this sentences.....
Response: We thank the reviewer for pointing this out. The sentence has been re-phrased properly (page 16-17 line 502-510).
- Lines 415-416: the fibro-chondrogenesis is not only characterized y Col1a1, nor the col1a1 is a specific marker of fibro-chondrogenesis. Please, re-phrase.
Response: We thank the reviewer for pointing this out. The reviewer is correct, however the results of the collagen I, of which 1a1 is a subunit of the larger collagen I protein, must be taken in conjunction with that of the collagen type II results. The presence of collagen II with I are indicative for a fibrochondrogensis and we have adjusted this accordingly in the manuscript to better reflect this (page 17 line 514-517).
- Line 420: 90-95
Response: We thank the reviewer for pointing this out. Numbers with integers at the beginning of a scientific sentence are normally written out. This was done in accordance to this rule.
- Lines 423-424: again, since authors did not explore the specific markers of articular cartilage, I would say chondrogenesis, and not articular chondrogenesis, please see article Hissnauer et al., 2010.
Response: We thank the reviewer for pointing this out. The material has been adjusted accordingly to better reflect this now. (page 17 line 524)
- Lines 433-435: If authors talk about fibro-chondrogenesis, it seems clear that this process here described has been identified and characterized by other researchers and pathologists before. Please, re-phrase it accordingly.
Response: We thank the reviewer for pointing this out. The material has been adjusted accordingly to better reflect this now.
- Lines 468-470: it is unclear whether author only evaluate the gene expression of reference genes in fresh muscle, or in all sampling times (7, 14 and 30 days). Please specify it.
Response: The reviewer is directed again to read the publications by Bustin et al 2009, 11 and 13. Reference genes chosen for use in this experiment were determined utilising ALL samples using geNorm program. Normalisation occurred over an endogenous control which was the fresh muscle tissue. There are other possibilities what can be normalised to but this the most appropriate form of doing so. As such the material in the text properly reflects present methodological standards in terms of the MIQE guidelines where the reviewer is directed to familiarise himself with the new nomenclature of the method.
- Lines 533-534: please, indicate the values of the 260/280 and 230/260 ratios to show the purity of the mRNA samples after nucleic acid isolation.
Response: We thank the reviewer for pointing this out. Given the vast raw data it would not be feasible to include this within the present manuscript. Given that 288 samples were measured we have included a range for the 260/280 values to show purity in the relevant materials and method section (page 19 line 635-638). The A230/260 was not measured as no DNA was extracted.
- Line 560: please, indicate the pH at which the alcian blue has been done since the type of proteoglycans stained might be different depending the ph.
Response: We thank the reviewer for pointing this out. The relevant ph at which the method was performed has now been included in the relevant materials and methods section (page 20 line 668-674)
- Conclusions: base on the lack of specific markers for articular chondrogenesis, present results suggest a kind of chondrogenesis, but it could be osteo- and/or fibro-chondrogenesis. Authors base on the presented data cannot conclude if this is fibro- or articular-chondrogenesis.
- Line 607: authors only used one specific type of tissue, muscle. Please, re-phrase it.
- Conclusions are a little bit vague. Can authors conclude something more specific, for instance which the combination used of morphogens were the most successful to induce a chondrogenesis process?
Response: We thank the reviewer for pointing this out. The conclusion has been adjusted appropriately to take all these criteria into consideration including certain parts of the abstract (page 21 line 716-726).
Reviewer 2 Report
Overall comment: This manuscript presents the role of members of the TGFb superfamily on the "transdifferentiation" of rat skeletal muscle towards chondrogenic lineage. The experiments were well designed and conducted, and the topic is of high importance. Nevertheless, there are issues to be considered.
Major comments:
- The authors should analyse markers (genes and proteins) of the muscular lineage. At the moment, it is shown the upregulation of chondrogenic markers, but it is not known if that is at the cost of downregulation of muscular markers, or if it is an independent process. This analysis would help to understand what is actually happening in the tissue. Is it "transdifferentiating" into a chondrogenic tissue, or are there just some progenitor cells that instead of following the muscular default are being stimulated towards chondrogenesis?
- Throughout the text it is given the impression that a regimen in which BMP2 is administered on the first 7 days, and afterwards TGFb + OP1, would be the most suitable to induce chondrogenesis in the system. However, it seems this is just a hypothesis based on the observation that BMP2 alone greater induced chondrogenesis until day 7, while afterwards the groups treated with TGFb + OP1 performed better. A regimen of BMP2 until day 7 + TGFb and OP1 afterwards have not been experimentally tested exactly. This should be made clearer. Since, that hypothesis could only be proved experimentally.
Minor comments:
. Line 73 - "transmute" should be replaced by another term. Are the authors trying to say "transdifferentiate"? Otherwise, could the authors better explain what they mean by "transmute"?
. Line 89 - Can the authors explain why 48h was the chosen timepoint for the single application system?
. It is not clear in the graphs, both bar chart and lines whether statistical analysis is been reffered as treatment vs control only, or single application vs singular application? Actually, I would suggest a more in depth statistical analysis between groups (not only against control), in order to better evidence a group superiority over another.
. The Figures for Col1, Col10 and ALP should be presented in the main text, and not as supplemental, unless there is a maximum number of figures allowed. They are important for overall results. Tables can remain supplemental.
. Fig 4 could be much improved. Instead of A1, B1, A2, B2 etc it would be much easier to the reader if the treatment was actually written on the image. Is there a negative control for the immunohistochemistry? The authors state the positive is in green, but it is hard to observe that. Possibly zoomed in images could help on visualisation. This could also be applied for the alcian blue images. Although the blue is easier to observe, zoomed in images would help to verify what actually is being positive stained.
.Line 420 - orthographic error in "Ninty".
. Line 423 - articular should be replaced by hyaline.
. Section 4.4 - can the authors state in which pH the alcian blue was used? Also, what was the counterstaining? Also, can the authors better describe the quantification performed in imageJ? What was considered positive alcian blue staining? How was that selected in the software?
. I would like to suggest some references, which I believe could enrich the discussion, especially on the chondrogenesis vs myogenic perspective:
https://www.ncbi.nlm.nih.gov/pmc/articles/PMC4297645/ - mouse skeletal stem cell - no myogenic differentiation
https://pubmed.ncbi.nlm.nih.gov/15776485/?dopt=Abstract
https://pubmed.ncbi.nlm.nih.gov/15776485/?dopt=Abstract - Examples of vascular muscle transdifferentiation towards cartilage
https://www.ncbi.nlm.nih.gov/pmc/articles/PMC3388093/
https://www.sciencedirect.com/science/article/pii/S1063458408001350 - chondrogenic differentiation of muscle progenitor cells
Author Response
Overall response to reviewers’ comments
Overall comment: This manuscript presents the role of members of the TGFb superfamily on the "transdifferentiation" of rat skeletal muscle towards chondrogenic lineage. The experiments were well designed and conducted, and the topic is of high importance. Nevertheless, there are issues to be considered.
Response: We would like to thank the reviewer for his/her appraisal of the work and thank him/her for pointing out critical aspects to make this manuscript suitable for publication within the Journal. Most of all the comments have been addressed that the reviewer has suggested including improving the quality of the Scientific English of the manuscript.
MAJOR COMMENTS
- The authors should analyse markers (genes and proteins) of the muscular lineage. At the moment, it is shown the upregulation of chondrogenic markers, but it is not known if that is at the cost of downregulation of muscular markers, or if it is an independent process. This analysis would help to understand what is actually happening in the tissue. Is it "transdifferentiating" into a chondrogenic tissue, or are there just some progenitor cells that instead of following the muscular default are being stimulated towards chondrogenesis?
Response: We thank the reviewer for their comment. As this is truly just at present a new model in which we wanted to see what would happen the aspect of then further assessing with more genes and other markers is something we will/are presently investigated in other studies of a similar nature that will form part of a series of systematic studies looking gradually in greater detail at what is happening here. Given that a nearly indefinite amount of variables have to be considered and relevant assays let alone projects we thank the reviewer for providing us with key aspects that need to be incorporated in all future studies of this nature.
- Throughout the text it is given the impression that a regimen in which BMP2 is administered on the first 7 days, and afterwards TGFb + OP1, would be the most suitable to induce chondrogenesis in the system. However, it seems this is just a hypothesis based on the observation that BMP2 alone greater induced chondrogenesis until day 7, while afterwards the groups treated with TGFb + OP1 performed better. A regimen of BMP2 until day 7 + TGFb and OP1 afterwards have not been experimentally tested exactly. This should be made clearer. Since, that hypothesis could only be proved experimentally.
Response: We thank the reviewer for pointing this out. We have gone through the text again and adjusted it to better reflect what the study is showing. Abstract, parts of the introduction and discussion have been adjusted in this regard.
MINOR COMMENTS
- Line 73 - "transmute" should be replaced by another term. Are the authors trying to say "transdifferentiate"? Otherwise, could the authors better explain what they mean by "transmute"?
Response: Transmutation derives from Physics and is defined as the changing of one element into another by radioactive decay, nuclear bombardment, or similar processes OR in Biology as changing of one species into another. We propose this term in our manuscript similarly to physics that governs all biological processes being the principle foundation of our field, we are changing the tissue into something else never before attempted using corresponding signals. This better reflects also the message of the manuscript as in our opinion, given more research into what we have attempted her and as we highlighted in our discussion and conclusion, it is our believe that any tissue can be changed into another tissue type soley dependant on the signals.
- line 89 - Can the authors explain why 48h was the chosen timepoint for the single application system?
Response: We chose 48h simply over 24h or 72h to see if this would be sufficient to induce a tissue wide response as is normally observed in vivo using similar time periods related to morphogen half-life and reported by diverse research projects. Ultimately, this will also need to be expanded on in relation to the comment 2 of the major comments, in which it is relevant to know how long a given signals needs to be active and then be removed or supplemented with another signal. This is again for future studies to investigate further.
- It is not clear in the graphs, both bar chart and lines whether statistical analysis is been reffered as treatment vs control only, or single application vs singular application? Actually, I would suggest a more in depth statistical analysis between groups (not only against control), in order to better evidence a group superiority over another.
Response: We thank the reviewer for pointing this out in which we have slightly tweaked the various sections to properly reflect what statistical evaluations were performed and between which groups. The material had been presented in great detail beforehand in which we believe the reviewer simply missed this as it was not clear enough. For the reviewers’ convenience, in each of the figures, part (A) presented the comparison between control group (normal medium) and the different morphogens treatments with significant differences represented by asterisks. The depth of the statistical analysis comparing the mean of each group and the mean of every other group using a one-way ANOVA with Tukey's multiple comparisons were performed and presented in the supplementary material. The asterisks in part (B) represent the significance of difference between single application and continuous application, while the comparisons among different time points were also performed and presented in the supplementary.
- The Figures for Col1, Col10 and ALP should be presented in the main text, and not as supplemental, unless there is a maximum number of figures allowed. They are important for overall results. Tables can remain supplemental.
Response: We thank the reviewer for indicating this to us. We have informed ourselves with the journals policies in terms of maximum figures permitted where we are able to incorporate the figures for Col1, Col10 and ALP as we presented in the manuscript.
- Fig 4 could be much improved. Instead of A1, B1, A2, B2 etc it would be much easier to the reader if the treatment was actually written on the image. Is there a negative control for the immunohistochemistry? The authors state the positive is in green, but it is hard to observe that. Possibly zoomed in images could help on visualisation. This could also be applied for the alcian blue images. Although the blue is easier to observe, zoomed in images would help to verify what actually is being positive stained.
Response: We thank the reviewer for their comment here. The Figure has been adjusted accordingly with specific zoomed in areas highlighting what the reader should be looking, including colors, at with treatment types having been placed appropriately that we hope meet with the reviewers approval (page 11 line 314-322). For the negative controls these were established by using the antibody diluent without any primary aggrecan antibody, which we have included in the relevant materials and methods section (page 21 line 685-686).
- Line 420 - orthographic error in "Ninty".
Response: We thank the reviewer for pointing this out. Numbers with integers at the beginning of a scientific sentence are normally written out. This was done in accordance to this rule.
- Line 423 - articular should be replaced by hyaline.
Response: We would like to than the reviewer for pointing this out to us. We have adjusted the wording appropriately where relevant (page 17 line 532-533)
- Section 4.4 - can the authors state in which pH the alcian blue was used? Also, what was the counterstaining? Also, can the authors better describe the quantification performed in imageJ? What was considered positive alcian blue staining? How was that selected in the software?
Response: We thank the reviewer for their comment. As per a similar request by reviewer 1 we have now added all the relevant information and provided a range where relevant page 20 line 668-674.
Pertaining to the reviewer’s second comment on the ImageJ, we believe this is not relevant to be included within the manuscript as this pertains to our own specific criteria for positive area selection. This can vary greatly between experiments and as such deviates accordingly. In the presented manuscript, as such for the reviewers own information, we used the specific RGB ranges to select the target color in histogram-based mode to reduce the variance in the selection of the positive area of the different figures. The RGB ranges of our positive area were R: 113-233, G: 156-244 and B: 195-245 accordingly.
- I would like to suggest some references, which I believe could enrich the discussion, especially on the chondrogenesis vs myogenic perspective:
https://www.ncbi.nlm.nih.gov/pmc/articles/PMC4297645/ - mouse skeletal stem cell - no myogenic differentiation
https://pubmed.ncbi.nlm.nih.gov/15776485/?dopt=Abstract
https://pubmed.ncbi.nlm.nih.gov/15776485/?dopt=Abstract - Examples of vascular muscle transdifferentiation towards cartilage
https://www.ncbi.nlm.nih.gov/pmc/articles/PMC3388093/
https://www.sciencedirect.com/science/article/pii/S1063458408001350 - chondrogenic differentiation of muscle progenitor cells
Response: We would like to thank the reviewer for assisting us with extra references that we have incorporated into our discussion where we have briefly expanded on some aspect but without negating the central message that we are trying to bring across (page 17 line 534-557). Whilst the chondrogenesis vs myogenic perspective is of relevance we do believe that in our follow-up studies this would be far better suited to be discussed as here we are incorporating also cellular based research and may be more relevant to discuss here.
Reviewer 3 Report
The authors determined the effects of BMP-2, TGF-β3 and/or BMP-7 treatment on the chondrogenesis of muscle tissue. Although the results are not easy to be applied in clinical directly, this is an interesting topic and design. The authors completed appropriate results, discussion, and conclusion. Moreover, the limitations of experimental designs were also described in the Discussion. It is well-written, scientifically sound and I don't have others comment.
The study described and compared the single and combined treatment effects of three members of TGF-b superfamily, i.e., BMP2, TGFb3, and BMP7, on the chondrogenesis of muscle tissues. Their data showed that different treating modes of these three growth factors could temporally and synergistically regulate the chondrogenic marker gene expression, including aggrecan, different types of collagens, SOX9, and ALP, in muscle tissue. One of the aims in this study was to elucidate the temporal signaling sequence during tissue morphogenesis. Moreover, on the tissue engineering opinion, the authors also demonstrated that the appropriate combinations of various types of morphogens and appropriate treating periods might change the tissue morphology and pattern.
Comment (1): The manuscript writing should be more condensed and streamlined, especially in Introduction. Moreover, the purpose of using muscle tissue in the study needs more elucidation in the Introduction (this is clearer in Abstract and Conclusion, but not in Introduction).
Comment (2): It is not easy to know the exact level of gene expression in the bar graph in some figures, e.g., Fig. 1A, because of the Y-axis definition. Moreover, some of the error bar are too high to know the exact values.
Author Response
Please see the attachement.

Reviewer 4 Report
This is a manuscript taking the first steps in a highly ambitious long-term project to build an ectopic cartilage model for use in studying and optimizing tissue engineering protocols based on the temporally prescribed application of multiple morphogens. This novel high risk, high reward model has the potential to be viewed as highly controversial. Nevertheless, the introduction is well-reasoned, the experiments are generally well-designed, the analysis of the data is robust, the conclusions are justified by the data, and the limitations of the data are clearly presented. Almost all of the questions this reviewer had while reading the manuscript were eventually answered later in the text. The largest weakness of the experimental design is the exclusive reliance on gene expression analysis for aggrecan, SOX9, and collagens I, II, and X. Immunohistochemical analysis of these markers would not only strengthen the manuscript overall but may also clarify the question posed on lines 543-544 regarding the potential peaking of expression levels between time points. The addition of these IHC data would be ideal, but is not viewed as necessary, provided this limitation is acknowledged in the discussion.
Author Response
Please see the attachement.
Round 2
Reviewer 1 Report
In the present version of the manuscript with reference IJMS-815627 and entitled: “Temporal TGF-β supergene family signalling cues modulating tissue morphogenesis: chondrogenesis within a muscle tissue model?”, authors have considered all my suggestions/comments performed on the previous version. Reviewer regrets that most of the suggestions/recommendations made to improve the clarity of the results here presented and the additional analysis that will support the hypothesis made by authors have been turned down, always arguing that the recommended analysis will be applied to future research studies, thus recognizing their value and need for supporting author hypothesis. Reviewer can understand that authors might not have the expertise equipment or samples to perform all the suggested analysis (eg protein phosphorylation), but denying to perform at least one gene quantification of the markers for articular cartilage (when authors performed the analysis of gene expression in 288 samples,…. I guess the authors should have at least some leftover cDNA from these samples to test whether a clearly better maker is expressed or not. Moreover, a hierarchical clustering do not imply the collection of further samples, just using a simple software to get further information. Based on the lecture of the new version and the responses to particular suggestions, this reviewer consider:
1-Introduction. It still lacks a proper state of the art and a clear justification why using a piece of muscle tissue (composed by different cell types, including stem cells) is a better approach to investigate the gene expression profile under different combinations of morphogens. Still, the disadvantage and benefits of using this approach versus a stem cell population is not clearly present to the readers. In this sense, authors stated to this request that introduction has been “briefly expanded” and added two references (27-28), but almost not reflecting the great advance and the benefit of using stem cell populations to the goal here proposed. A quick literature search provides an immense list of articles dealing with cartilage engineering using stem cell populations. Please, provide a clear statement whether the experimental approach here used present some benefits over the use of a stem cell population. In this way, readers might be able to take their own conclusions about the present piece of research regarding the suitability of the experimental approach used to respond the proposed hypothesis.
2-Authors have been requested to provide information of the different genes that were evaluated by qPCR analysis, not to list them in the introduction. Please, provide a brief description why each of them is related to osteo/chondrogenesis.
3-Major drawback is still present in the present version. As stated in the previous review, authors need to evaluate (at least 2) genes that were reported as good biomarkers of articular chondrogenesis (from Hissnauer et al., 2010; https://doi.org/10.1016/j.joca.2010.10.002) in order to support their hypothesis. Again, I kindly request authors to evaluate them. Although the genes evaluated by authors are still fundamental markers of chondrogenesis and/or osteogenesis, they are not valid for informing about the type of chondrogenesis is taking plase. This analysis is needed to sustain authors hypothesis (lines 522-526: in all likelihood the muscle tissue was stimulated and tending towards a specific type chondrogenesis that could have been articular when stimulated by the selected growth factors of TGF-β superfamily applied continuously, alone or in combinations. ) about the type of chondrogenesis is formed (articular, fibrochondogenesis,…). In fact, although classical markers (here used) can inform a tissue undergoes chondrogenesis (as authors stated in the text), it doesn’t support if it is actually articular chondrogenesis.
4-Reviewer regrets that authors do not found a better way to represent gene expression data on their figures, since the clarity of the results would be clearly improved as well as reducing the repetition of data. In fact, authors should not represent in different figures with the same data. In this sense, hierarchical clustering would provide to authors with a wider vision of their results and can identify which experimental groups respond in the same way in an easy manner. In fact, authors can do it even compiling data from different modalities and stimulation duration at each sampling time.
5-As shown by the micrographs included in the manuscript, there is a specific region of the tissue section that evolves towards a chondrogenic state, but not all the region. This is why authors should “defense” their experimental approach over the use of stem cell populations, and why a kind of analysis on cell differentiation and proliferation should be conducted.
6-Last suggestion to authors, please stick your hypothesis and conclussions to what is shown by your results. In addition to the issue of being directed part of the cell population to chondrogenesis (if articular or not), authors cannot conclude that “muscle tissue could be “transmuted” ex vivo into any tissue type”. No other type of tissue rather than chondrogenic has been induced by morphogens here used.
Author Response
Response to Reviewers Comments
Reviewer 1
Overall response to reviewers’ comments
In the present version of the manuscript with reference IJMS-815627 and entitled: “Temporal TGF-β supergene family signalling cues modulating tissue morphogenesis: chondrogenesis within a muscle tissue model?”, authors have considered all my suggestions/comments performed on the previous version. Reviewer regrets that most of the suggestions/recommendations made to improve the clarity of the results here presented and the additional analysis that will support the hypothesis made by authors have been turned down, always arguing that the recommended analysis will be applied to future research studies, thus recognizing their value and need for supporting author hypothesis. Reviewer can understand that authors might not have the expertise equipment or samples to perform all the suggested analysis (eg protein phosphorylation), but denying to perform at least one gene quantification of the markers for articular cartilage (when authors performed the analysis of gene expression in 288 samples,…. I guess the authors should have at least some leftover cDNA from these samples to test whether a clearly better maker is expressed or not. Moreover, a hierarchical clustering do not imply the collection of further samples, just using a simple software to get further information. Based on the lecture of the new version and the responses to particular suggestions, this reviewer consider:
Response: We would like to thank the reviewer for his/her suggestions to further improve the manuscript with additional tests, however we reject the insinuating remarks by the reviewer suggesting that we are arguing and have ignored “most” of his/her comments. The reviewer is kindly asked to please refrain from making future assumptions of other laboratories or research groups or “guessing” if we have something or not. The reviewer is not present in the research group or laboratory. As such, I as the corresponding and senior author have the following to response after careful consideration in light of the comments of the reviewer:
- I would welcome doing additional test if the material were available to accommodate the reviewer’s request. I run a tight program where all steps have been fully optimized to maximize experiments, funding and time. Sufficient cDNA was produced to accommodate the genes we had chosen. However, as per the last response the mRNA is no longer viable to test further genes and whilst there is still “some” cDNA, as the reviewer has unprofessionally assumed, the quantity is insufficient such that it is not 1:1 with our methodological setup and will generate data gaps essential for proper qPCR results. However, I agree that these additional genes would strengthen the results. To reach a consensus with the reviewer the material has been adjusted to reflect appropriately what the results are showing indicating where relevant what it could have become had the relevant additional genes been included as this is still relevant.
- I thank the reviewer for suggesting to us use of hierarchical clustering to present our gene expression data. I misunderstood previously what the reviewer was getting at. We performed the hierarchical clustering and generated the heatmap in RStudio (Boston, MA, USA). As per the reviewers suggestion, we compiled data from different modalities and stimulation duration at each sampling time. We have 6 replicates in each group (15) which means at each time point, we analysed 90 samples. And finally, we got 3 hierarchical clustering figures. At day 7, we found the gene expression in the group applied rBMP-2 continuously was different from the others. At day 14 and 30, the figures help us identify the continuous application groups respond in the same way and differ from the single application and control groups (xFigure1-3).
Although hierarchical clustering does reduce the repetition of data, and provided us with a wider vision of our results that can identify which experimental groups respond in the same way in an easy manner, the information that we obtained from the hierarchical clustering was still limited. For example, we cannot figure out which groups were significantly different from the control group at each time point, and the comparison between each sampling time cannot be performed. Additionally, due to a large number of our experimental groups and small replicate range, to find the exact way each group responded is difficult. Furthermore, the information we got from the hierarchical clustering can also be obtained from our original figure although here the problem of data redundancy existed. For example, the dominant position of rBMP-2 at day 7 and the difference between continuous and single stimulation, proved necessary such that the two parts of each picture exist together. Furthermore, we tried to find a parameter that reflected the value of a certain group, such as the mean value, but this violated the principle of hierarchical clustering. Subsequently, if we did include the hierarchical clustering as evidenced in the attached three figures of our response here, the results in our opinion would still be limited and difficult to interpret thus in our opinion we would not include them and reviewer 2 was able understand our data as it was originally presented.
However, if the reviewer believes that these result of hierarchical clustering are irreplaceable and assist in the strengthening the logical interpretations, we can add these figures as part of the final version of the manuscript to be published or in supplementary materials section.
|
xigure 1. Hierarchical clustering of gene expression of all the samples at day 7. The dendrogram shows two main clusters of the group applied rBMP-2 continuously and the other groups. The distance measure used in clustering rows and column was euclidean, the agglomeration method to be used to cluster was average. Con.=Continuous, Sin.=Single, B= rBMP-2 treated group, T= rTGF-β3 treated group, O= rBMP-7 treated group, B+T = rBMP-2 + rTGF-β3 treated group, B+O=rBMP-2 + rBMP-7 treated group, T+O= rTGF-β3 + rBMP-7 treated group, B+T+O= rBMP-2 + rTGF-β3 + rBMP-7 treated group, ACAN= aggrecan, SOX9= sex-determining region Y (SRY)-box 9, Col2α1=collagen type II alpha 1, Col1α1=collagen type I alpha 1, Col10α1=collagen type X alpha 1, ALP=alkaline phosphatase. |
|
xFigure 2. Hierarchical clustering of gene expression of all the samples at day 14. The dendrogram shows two main clusters of the groups applied morphogens continuously and the groupa applied morphogens for only 48 hours (containing the control group). The distance measure used in clustering rows and column was euclidean, the agglomeration method to be used to cluster was average. Con.=Continuous, Sin.=Single, B= rBMP-2 treated group, T= rTGF-β3 treated group, O= rBMP-7 treated group, B+T = rBMP-2 + rTGF-β3 treated group, B+O=rBMP-2 + rBMP-7 treated group, T+O= rTGF-β3 + rBMP-7 treated group, B+T+O= rBMP-2 + rTGF-β3 + rBMP-7 treated group, ACAN= aggrecan, SOX9= sex-determining region Y (SRY)-box 9, Col2α1=collagen type II alpha 1, Col1α1=collagen type I alpha 1, Col10α1=collagen type X alpha 1, ALP=alkaline phosphatase. |
|
xFigure 3. Hierarchical clustering of gene expression of all the samples at day 30. The dendrogram shows two main clusters of the groups applied morphogens continuously and the groupa applied morphogens for only 48 hours (containing the control group). The distance measure used in clustering rows and column was euclidean, the agglomeration method to be used to cluster was average. Con.=Continuous, Sin.=Single, B= rBMP-2 treated group, T= rTGF-β3 treated group, O= rBMP-7 treated group, B+T = rBMP-2 + rTGF-β3 treated group, B+O=rBMP-2 + rBMP-7 treated group, T+O= rTGF-β3 + rBMP-7 treated group, B+T+O= rBMP-2 + rTGF-β3 + rBMP-7 treated group, ACAN= aggrecan, SOX9= sex-determining region Y (SRY)-box 9, Col2α1=collagen type II alpha 1, Col1α1=collagen type I alpha 1, Col10α1=collagen type X alpha 1, ALP=alkaline phosphatase. |
Point-by-point response:
1-Introduction. It still lacks a proper state of the art and a clear justification why using a piece of muscle tissue (composed by different cell types, including stem cells) is a better approach to investigate the gene expression profile under different combinations of morphogens. Still, the disadvantage and benefits of using this approach versus a stem cell population is not clearly present to the readers. In this sense, authors stated to this request that introduction has been “briefly expanded” and added two references (27-28), but almost not reflecting the great advance and the benefit of using stem cell populations to the goal here proposed. A quick literature search provides an immense list of articles dealing with cartilage engineering using stem cell populations. Please, provide a clear statement whether the experimental approach here used present some benefits over the use of a stem cell population. In this way, readers might be able to take their own conclusions about the present piece of research regarding the suitability of the experimental approach used to respond the proposed hypothesis.
5-As shown by the micrographs included in the manuscript, there is a specific region of the tissue section that evolves towards a chondrogenic state, but not all the region. This is why authors should “defense” their experimental approach over the use of stem cell populations, and why a kind of analysis on cell differentiation and proliferation should be conducted.
Response: We thank the reviewer for his/her comment. We added a new reference which performed a similar study but treated mesenchymal stem cells as the object. Considering the previous description of cell experiments, we thought it was enough to show the current situation of stem cells in TGF superfamily research in the introduction section, after all, the introduction is not meant to be a review. In addition, we also added the shortage of cell-based study. We are not afraid to show the great advance and the benefit of using stem cell populations in basic experiments. As the reviewer said, “ A quick literature search provides an immense list of articles dealing with “cartilage engineering” using stem cell populations.”, which was why we pay attention to the lack of direct research on tissue culture. Therefore, we attempted to propose a new model of muscle tissue culture and try to explore its response to TGF-β superfamily. From an objective point of view, we couldn't and didn't need to evaluate and compare the advantages and disadvantages of the cell and tissue in this article, but we thank the reviewers for providing us with a fantastic experimental idea, that is, to compare the cell culture and tissue culture through multiple ways.
The sections have been tweaked accordingly (page 2 lines 74-85).
2-Authors have been requested to provide information of the different genes that were evaluated by qPCR analysis, not to list them in the introduction. Please, provide a brief description why each of them is related to osteo/chondrogenesis.
Response: We thank the reviewer for their comment. The relevant section has been adjusted to properly reflect this now and how each gene is involved in the processes. (page 3 lines 98-113)
3-Major drawback is still present in the present version. As stated in the previous review, authors need to evaluate (at least 2) genes that were reported as good biomarkers of articular chondrogenesis (from Hissnauer et al., 2010; https://doi.org/10.1016/j.joca.2010.10.002) in order to support their hypothesis. Again, I kindly request authors to evaluate them. Although the genes evaluated by authors are still fundamental markers of chondrogenesis and/or osteogenesis, they are not valid for informing about the type of chondrogenesis is taking plase. This analysis is needed to sustain authors hypothesis (lines 522-526: in all likelihood the muscle tissue was stimulated and tending towards a specific type chondrogenesis that could have been articular when stimulated by the selected growth factors of TGF-β superfamily applied continuously, alone or in combinations. ) about the type of chondrogenesis is formed (articular, fibrochondogenesis,…). In fact, although classical markers (here used) can inform a tissue undergoes chondrogenesis (as authors stated in the text), it doesn’t support if it is actually articular chondrogenesis.
Response: We thank the reviewer for his/her comment again on this issue. We direct the reviewer under our overall response 1. Where relevant in the manuscript text has been adjusted to reflect actual results now where relevant (page 18 lines 552-557).
4-Reviewer regrets that authors do not found a better way to represent gene expression data on their figures, since the clarity of the results would be clearly improved as well as reducing the repetition of data. In fact, authors should not represent in different figures with the same data. In this sense, hierarchical clustering would provide to authors with a wider vision of their results and can identify which experimental groups respond in the same way in an easy manner. In fact, authors can do it even compiling data from different modalities and stimulation duration at each sampling time.
Response: We thank the reviewer for his/her comment again on this issue. We direct the reviewer under our overall response 2.
6-Last suggestion to authors, please stick your hypothesis and conclussions to what is shown by your results. In addition to the issue of being directed part of the cell population to chondrogenesis (if articular or not), authors cannot conclude that “muscle tissue could be “transmuted” ex vivo into any tissue type”. No other type of tissue rather than chondrogenic has been induced by morphogens here used.
Response: We thank the reviewer for his/her comment. We agree with the reviewer and have adjusted the material as required (page 22 lines 759-760).

Reviewer 2 Report
I would like to thank the authors for addressing my comments. Nevertheless, I would still like to discuss some minor issues, that could possibly be addressed by the authors, in order to further improve the quality of the manuscript. In order to facilitate the process, I am repeating the previous comments, adding current additional comments based on authors' responses.
MINOR COMMENTS
- Line 73 - "transmute" should be replaced by another term. Are the authors trying to say "transdifferentiate"? Otherwise, could the authors better explain what they mean by "transmute"?
Response: Transmutation derives from Physics and is defined as the changing of one element into another by radioactive decay, nuclear bombardment, or similar processes OR in Biology as changing of one species into another. We propose this term in our manuscript similarly to physics that governs all biological processes being the principle foundation of our field, we are changing the tissue into something else never before attempted using corresponding signals. This better reflects also the message of the manuscript as in our opinion, given more research into what we have attempted her and as we highlighted in our discussion and conclusion, it is our believe that any tissue can be changed into another tissue type soley dependant on the signals.
Additional comment: I thank the authors explanation on transmutation in Physics and Biology and respect their willing to keep with the term "transmute". However, I believe what the authors are stating as "transmutation" is already well-accepted as "transdifferentiation", which would be my suggestion as alternative term for better acceptance and understanding of the idea proposed in the manuscript. Anyway, again, if the authors want to keep the term "transmute", please feel free to go on.
- It is not clear in the graphs, both bar chart and lines whether statistical analysis is been reffered as treatment vs control only, or single application vs singular application? Actually, I would suggest a more in depth statistical analysis between groups (not only against control), in order to better evidence a group superiority over another.
Response: We thank the reviewer for pointing this out in which we have slightly tweaked the various sections to properly reflect what statistical evaluations were performed and between which groups. The material had been presented in great detail beforehand in which we believe the reviewer simply missed this as it was not clear enough. For the reviewers’ convenience, in each of the figures, part (A) presented the comparison between control group (normal medium) and the different morphogens treatments with significant differences represented by asterisks. The depth of the statistical analysis comparing the mean of each group and the mean of every other group using a one-way ANOVA with Tukey's multiple comparisons were performed and presented in the supplementary material. The asterisks in part (B) represent the significance of difference between single application and continuous application, while the comparisons among different time points were also performed and presented in the supplementary.
Additional comment: I thank the authors for the explanation. Again, just thinking on facilitating the reader's life, I would suggest that such explanation would be briefly given in the figure legends. If I may suggest, I would add something like: "* P<0.05, ** P<0.01, *** P<0.001, **** P<0.0001. In (A) statistical significance is expressed between each group and the control group (normal medium), while in (B) it is expressed in each time point between continuous stimulation and single stimulation."
- Section 4.4 - can the authors state in which pH the alcian blue was used? Also, what was the counterstaining? Also, can the authors better describe the quantification performed in imageJ? What was considered positive alcian blue staining? How was that selected in the software?
Response: We thank the reviewer for their comment. As per a similar request by reviewer 1 we have now added all the relevant information and provided a range where relevant page 20 line 668-674.
Pertaining to the reviewer’s second comment on the ImageJ, we believe this is not relevant to be included within the manuscript as this pertains to our own specific criteria for positive area selection. This can vary greatly between experiments and as such deviates accordingly. In the presented manuscript, as such for the reviewers own information, we used the specific RGB ranges to select the target color in histogram-based mode to reduce the variance in the selection of the positive area of the different figures. The RGB ranges of our positive area were R: 113-233, G: 156-244 and B: 195-245 accordingly.
Additional comment: The authors stated to have used Alcian Blue pH 2.5 (line 671), and that this stains acidic polysaccharides (line 320). Alcian Blue pH 1.0 vs pH 2.5 can be very tricky. At lower pH (pH1.0) it will stain most sulfated and highly acidic mucins, as for chondroitin sulphate, while pH 2.5 will stain more carboxylated less acidic mucins, which indeed can be found in cartilage, but also generally in connective tissue. For instance, this paper present positive alcian blue pH 2.5 staining in decellularized muscle samples (https://www.mdpi.com/1422-0067/16/7/14808/htm). It is understandable that all the other results suggest chondrogenic differentiation, and not a general connective tissue accumulation. Nevertheless, I would suggest the authors to change "acidic polysaccharides" in line 320 to "low acidic/carboxylated polysaccharides". In regards to ImageJ discussion, in an attempt to achieve greater reproducibility in science, I kindly disagree with the authors. Since there can be great variation among experiments and labs in performing this quantification, I would recommend that the RGB range for positivity used by the authors is actually disclaimed in the manuscript. With this, other researchers will be able to use similar parameters, or not, if they do not judge them feasible for their scenario.
Author Response
Reviewer 2
Overall response to reviewers’ comments
I would like to thank the authors for addressing my comments. Nevertheless, I would still like to discuss some minor issues that could possibly be addressed by the authors, in order to further improve the quality of the manuscript. In order to facilitate the process, I am repeating the previous comments, adding current additional comments based on authors' responses.
Response: We thank the reviewer for positive appraisal of our previous corrections in which we have considered all of his/her additional minor comments to help make the manuscript ready for publication.
MINOR COMMENTS
- Line 73 - "transmute" should be replaced by another term. Are the authors trying to say "transdifferentiate"? Otherwise, could the authors better explain what they mean by "transmute"? Response: Transmutation derives from Physics and is defined as the changing of one element into another by radioactive decay, nuclear bombardment, or similar processes OR in Biology as changing of one species into another. We propose this term in our manuscript similarly to physics that governs all biological processes being the principle foundation of our field, we are changing the tissue into something else never before attempted using corresponding signals. This better reflects also the message of the manuscript as in our opinion, given more research into what we have attempted her and as we highlighted in our discussion and conclusion, it is our believe that any tissue can be changed into another tissue type soley dependant on the signals.
Additional comment: I thank the authors explanation on transmutation in Physics and Biology and respect their willing to keep with the term "transmute". However, I believe what the authors are stating as "transmutation" is already well-accepted as "transdifferentiation", which would be my suggestion as alternative term for better acceptance and understanding of the idea proposed in the manuscript. Anyway, again, if the authors want to keep the term "transmute", please feel free to go on.
Response: We thank the reviewer for his/her comment and clarifying the two terms. However, related to the comments by reviewer 1 and the fact that we did not test other tissues yet that would suggest that indeed tissue could be transdifferentiated we have rephrased the relevant sections to better reflect this.
- It is not clear in the graphs, both bar chart and lines whether statistical analysis is been reffered as treatment vs control only, or single application vs singular application? Actually, I would suggest a more in depth statistical analysis between groups (not only against control), in order to better evidence a group superiority over another.
Response: We thank the reviewer for pointing this out in which we have slightly tweaked the various sections to properly reflect what statistical evaluations were performed and between which groups. The material had been presented in great detail beforehand in which we believe the reviewer simply missed this as it was not clear enough. For the reviewers’ convenience, in each of the figures, part (A) presented the comparison between control group (normal medium) and the different morphogens treatments with significant differences represented by asterisks. The depth of the statistical analysis comparing the mean of each group and the mean of every other group using a one-way ANOVA with Tukey's multiple comparisons were performed and presented in the supplementary material. The asterisks in part (B) represent the significance of difference between single application and continuous application, while the comparisons among different time points were also performed and presented in the supplementary.
Additional comment: I thank the authors for the explanation. Again, just thinking on facilitating the reader's life, I would suggest that such explanation would be briefly given in the figure legends. If I may suggest, I would add something like: "* P<0.05, ** P<0.01, *** P<0.001, **** P<0.0001. In (A) statistical significance is expressed between each group and the control group (normal medium), while in (B) it is expressed in each time point between continuous stimulation and single stimulation."
Response: We thank the reviewer for his/her comment. This has now been incorporated now in the figure legends to help show this and make it easier to understand.
- Section 4.4 - can the authors state in which pH the alcian blue was used? Also, what was the counterstaining? Also, can the authors better describe the quantification performed in imageJ? What was considered positive alcian blue staining? How was that selected in the software?
Response: We thank the reviewer for their comment. As per a similar request by reviewer 1 we have now added all the relevant information and provided a range where relevant page 20 line 668-674.
Pertaining to the reviewer’s second comment on the ImageJ, we believe this is not relevant to be included within the manuscript as this pertains to our own specific criteria for positive area selection. This can vary greatly between experiments and as such deviates accordingly. In the presented manuscript, as such for the reviewers own information, we used the specific RGB ranges to select the target color in histogram-based mode to reduce the variance in the selection of the positive area of the different figures. The RGB ranges of our positive area were R: 113-233, G: 156-244 and B: 195-245 accordingly.
Additional comment: The authors stated to have used Alcian Blue pH 2.5 (line 671), and that this stains acidic polysaccharides (line 320). Alcian Blue pH 1.0 vs pH 2.5 can be very tricky. At lower pH (pH1.0) it will stain most sulfated and highly acidic mucins, as for chondroitin sulphate, while pH 2.5 will stain more carboxylated less acidic mucins, which indeed can be found in cartilage, but also generally in connective tissue. For instance, this paper present positive alcian blue pH 2.5 staining in decellularized muscle samples (https://www.mdpi.com/1422-0067/16/7/14808/htm). It is understandable that all the other results suggest chondrogenic differentiation, and not a general connective tissue accumulation. Nevertheless, I would suggest the authors to change "acidic polysaccharides" in line 320 to "low acidic/carboxylated polysaccharides". In regards to ImageJ discussion, in an attempt to achieve greater reproducibility in science, I kindly disagree with the authors. Since there can be great variation among experiments and labs in performing this quantification, I would recommend that the RGB range for positivity used by the authors is actually disclaimed in the manuscript. With this, other researchers will be able to use similar parameters, or not, if they do not judge them feasible for their scenario.
Response: We thank the reviewer for his/her comment. The relevant figure legends sections have been adjusted for specificity of Alcian Blue staining at that pH and the necessary criteria for histomorph parameters with ImageJ have now been included in the relevant material and methods section.

Round 3
Reviewer 1 Report
In the 3rd version of the manuscript with reference IJMS-815627 and entitled: “Temporal TGF-β supergene family signalling cues modulating tissue morphogenesis: chondrogenesis within a muscle tissue model?”, authors have accepted to explore some of the propositions to improve the manuscript.
Considering what is presented in the new version, the hierarchical clusterings and the responses from the authors to the previous comments, present reviewer consider:
1-I will not fall on personal descriptions about the professional conduct of any one here in this reviewing process, and request to the corresponding author to do not take my suggestions wrongly.
To my first request conducting additional qPCR analysis of very interesting genes that will clearly support their hypothesis (and not with other intentions), authors should state that no longer mRNA and/or cDNA is still available for conducting such analysis, rather than answering that they might consider to explore them in future studies. Reviewer is obviously not at the corresponding author’s lab (in such situation, there will be a conflict of interest on reviewing present manuscript) and thus, after a not clear statement why authors would not conduct such proposed qPCRs, I insisted in performing such analysis with no clues whether this mRNA and/or cDNA is still available or not. As a researcher, present reviewer also conducts gene expression analysis, and we always try to keep enough biological material in order to repeat and/or do additional qPCR experiments if needed (also considering that a reviewer might ask for exploring additional genes). Whatever, since no more information could be obtained from the gene expression profiling in the samples, authors should not hypothesize on the type of chondrogenesis that morphogens might stimulate and thus, any hypothesis on this issue should be removed from text.
2-Authors continue to affirm along the manuscript things that have not been shown, explored and demonstrated in the present manuscript. Although they have corrected their statements in the previous version regarding whether muscle tissue could be “transmuted” ex vivo into any tissue type or not, in the present version there are statements about things that were not analyzed/explored. For example, in the abstract (Lines 24-26), authors stated: “…. is a world-first attempt to better understand how multiple morphogens affect tissue morphogenesis with time in order to determine the chronological order of what signals have to be applied when, for how long and with which other signals to induce and maintain a desired tissue morphogenesis”. Authors did not explore the best (or any) chronological order of different morphogens neither when it should be applied. Authors just explored the application of different morphogens (alone or in combination), but not when a different chronological order was applied for all these morphogens.
3-Regarding Hierarchical clustering. I’m not totally satisfied with the presented hierarchical clustering. I would like to see the results integrating all the analyses performed, including the alcian blue staining. In this way, authors might have a clearer and objective clue on what is really relevant on the gene expression process along the 30 day of follow up and that might be explain the development of a chondrogenesis, and /or what each combination of morphogens is really inducing.
In addition, and in contrast to authors, present reviewer considers that the clusterings (although not including all the data analyzed in the present manuscript) are showing the most important things (even when they do not include all the data). In this sense, at 7 days, although two clearly differentiated clustering are shown (as authors stated on their response), it is clearly shown that this are based on the close related gene expression results of Sox9 and Acan, the two genes more related with the initial development of chondrogenesis, and the high expression of Col2 and Alp genes. At 14 days, there is also two clusters, but now all the continuous exposure to morphogens are clustered together, not only the ones treated with BMP2. This objectively suggest that all muscle samples under continuous treatment with morphogens (independently of their different combinations) respond in the same way, independently of their different start (when BMP2 treatment was shown different from all the rest). At this time, clustering reflects the gene expression of Col2, Acan, Sox9 and Alp, resulting from a more advanced chondrogenesis stage. Finally, at day 30 the 2 previously identified clusters are maintained. In fact, and interesting finding is that although not clustered independently, single exposure to BMP2 + rTGF-β3 + rBMP-7, and most samples from BMP2 + rBMP-7, are able to induce high expression of ALP, Col1 and Col10. Such finding would not be possible to be observed is authors do not conduct the hierarchical clustering previously requested.
In this sense, I kindly request to authors to conduct a hierarchical clustering not only including gene expression data, but also alcian blue staining. Authors can explore which clustering should they present (one global, including all the data at the three-sampling time or several clusterings). I really think that this procedure will help them to identify what is induced under the different experimental combinations here evaluated.
4-Furthermore, present reviewer disagrees on some of the statements and conclusions that the authors are taken from these hierarchical clusterings. Particularly:
a)The reviewer is not in agreement with the statement of “we cannot figure out which groups were significantly different from the control group at each time point”. Hierarchical clustering are showing what is really there on a global basis. If control group is clustered together with another experimental groups, then it means that based on the global results, the control group is not behaving differentially to these particular groups with which is clustered to.
b)Regarding the statement “and the comparison between each sampling time cannot be performed”, present reviewer considers that yes, such comparison can be done. For these authors should include all the data corresponding to all the sampling points (and identifying it with that day of sampling. In this sense, such global clustering would offer the possibility to see is at different sampling time different experimental groups behave similarly.
c)Reviewer is also in disagreement with the statement of “Additionally, due to a large number of our experimental groups and small replicate range, to find the exact way each group responded is difficult”. Hierarchical clustering can be done in a robust way with only three replicates from each biological group and can be done with hundreds to thousand analyzed parameters (it is largely applied to microarrays and RNA-Seq experiments, where the expression of thousand genes are evaluated at different time points). It is not difficult, and it is the most objective analytical process to extract conclusions and identify how different experimental groups behave through comparisons with all them at the same time. This type of analysis avoids any skewed interpretation of the data that can be obtained through interpreting isolated data.
d)“Furthermore, the information we got from the hierarchical clustering can also be obtained from our original figure although here the problem of data redundancy existed”. Not, you are not able since you don’t have all the data together and compared globally.
5-“it nonetheless strongly suggests that with the exact temporal chronological order of signaling ligands muscle tissue could be stimulated and maintained ex vivo to undergo chondrogenesis”. Not all the cells from the muscle tissue section exposed to morphogens were stimulated to chondrogenesis, just only some of them. Were the myocytes or were the stem cells the cells stimulated towards a chondrogenesis? This is also why present reviewer ask authors to present in the introduction section the limitations of using a muscle tissue section over a stem cell population.
Round 4
Reviewer 1 Report
In the 4th version of the manuscript with reference IJMS-815627 and entitled: “Temporal TGF-β supergene family signalling cues modulating tissue morphogenesis: chondrogenesis within a muscle tissue model?”, authors have accepted to perform and include the hierarchical clusterings suggested from the first round of reviewing process. Authors also modified the text in the introduction and discussion sections in order to accommodate the previous suggestions (regarding the speculation on the type of chondrogenesis). Reviewer really appreciates the effort made by authors in this sense.
Minor details to be corrected:
Line-79: “that” is repeated.
Author Response
Please see the attachement.
